# Body mass index and childhood symptoms of depression, anxiety, and attention-deficit hyperactivity disorder: A within-family Mendelian randomization study

Amanda M Hughes[1,2]*, Eleanor Sanderson[1,2], Tim Morris[1,2], Ziada Ayorech[3,4], Martin Tesli[5,6], Helga Ask[3,5], Ted Reichborn-Kjennerud[5,7], Ole A Andreassen[6,7], Per Magnus[8], Øyvind Helgeland[9], Stefan Johansson[10,11], Pål Njølstad[12,13], George Davey Smith[1,2], Alexandra Havdahl[1,2,3,4,5†], Laura D Howe[1,2†], Neil M Davies[1,2,14†]

[1]Medical Research Council Integrative Epidemiology Unit, University of Bristol, Bristol, United Kingdom; [2]Population Health Sciences, Bristol Medical School, University of Bristol, Barley House, Oakfield Grove, Bristol, United Kingdom; [3]PROMENTA Research Centre, Department of Psychology, University of Oslo, Oslo, Norway; [4]Nic Waals Institute, Lovisenberg Diaconal Hospital, Oslo, Norway; [5]Department of Mental Disorders, Norwegian Institute of Public Health, Oslo, Norway; [6]Norwegian Centre for Mental Disorders Research (NORMENT), Division of Mental Health and Addiction, Oslo University Hospital, Oslo, Norway; [7]Institute of Clinical Medicine, University of Oslo, Oslo, Norway; [8]Centre for Fertility and Health, Norwegian Institute of Public Health, Oslo, Norway; [9]Center for Diabetes Research, Department of Clinical Science, University of Bergen, Bergen, Norway; [10]Department of Clinical Science, University of Bergen, Bergen, Norway; [11]Department of Medical Genetics, Haukeland University Hospital, Bergen, Norway; [12]Mohn Center for Diabetes Precision Medicine, Department of Clinical Science, University of Bergen, Bergen, Norway; [13]Children and Youth Clinic, Haukeland University Hospital, Bergen, Norway; [14]K.G. Jebsen Center for Genetic Epidemiology, Department of Public Health and Nursing, NTNU, Norwegian University of Science and Technology, Høgskoleringen, Norway

*For correspondence: amanda.hughes@bristol.ac.uk

†These authors contributed equally to this work

## Abstract

**Background:** Higher BMI in childhood is associated with emotional and behavioural problems, but these associations may not be causal. Results of previous genetic studies imply causal effects but may reflect influence of demography and the family environment.

**Methods:** This study used data on 40,949 8-year-old children and their parents from the Norwegian Mother, Father and Child Cohort Study (MoBa) and Medical Birth Registry of Norway (MBRN). We investigated the impact of BMI on symptoms of depression, anxiety, and attention-deficit hyperactivity disorder (ADHD) at age 8. We applied within-family Mendelian randomization, which accounts for familial effects by controlling for parental genotype.

**Results:** Within-family Mendelian randomization estimates using genetic variants associated with BMI in adults suggested that a child's own BMI increased their depressive symptoms (per 5 kg/m$^2$

increase in BMI, beta = 0.26 S.D., CI = −0.01,0.52, p=0.06) and ADHD symptoms (beta = 0.38 S.D., CI = 0.09,0.63, p=0.009). These estimates also suggested maternal BMI, or related factors, may independently affect a child's depressive symptoms (per 5 kg/m$^2$ increase in maternal BMI, beta = 0.11 S.D., CI:0.02,0.09, p=0.01). However, within-family Mendelian randomization using genetic variants associated with retrospectively-reported childhood body size did not support an impact of BMI on these outcomes. There was little evidence from any estimate that the parents' BMI affected the child's ADHD symptoms, or that the child's or parents' BMI affected the child's anxiety symptoms.

**Conclusions:** We found inconsistent evidence that a child's BMI affected their depressive and ADHD symptoms, and little evidence that a child's BMI affected their anxiety symptoms. There was limited evidence of an influence of parents' BMI. Genetic studies in samples of unrelated individuals, or using genetic variants associated with adult BMI, may have overestimated the causal effects of a child's own BMI.

**Funding:** This research was funded by the Health Foundation. It is part of the HARVEST collaboration, supported by the Research Council of Norway. Individual co-author funding: the European Research Council, the South-Eastern Norway Regional Health Authority, the Research Council of Norway, Helse Vest, the Novo Nordisk Foundation, the University of Bergen, the South-Eastern Norway Regional Health Authority, the Trond Mohn Foundation, the Western Norway Regional Health Authority, the Norwegian Diabetes Association, the UK Medical Research Council. The Medical Research Council (MRC) and the University of Bristol support the MRC Integrative Epidemiology Unit.

## Editor's evaluation

The manuscript uses genetic effects on BMI to test whether BMI affecxts childhood emotional and behavioural problems: symptoms of depression, anxiety, and attention-deficit and hyperactivity disorder (ADHD) at age 8. By using a within-family design in a large sample of children with genotyped parents in Norway, the study finds that previous estimates of the effect of BMI on childhood emotional and behavioural symptoms may have been overestimated due to confounding with the environment. Larger samples will be needed to determine whether there is a causal effect of BMI on childhood emotional or behavioural problems, and what size it is.

## Introduction

Children with high body mass index (BMI) have been found to have greater risk of emotional and behavioural problems, including symptoms and diagnoses of depression (*Lindberg et al., 2020*; *Patalay and Hardman, 2019*; *Geoffroy et al., 2014*; *Quek et al., 2017*) anxiety (*Lindberg et al., 2020*) and attention-deficit hyperactivity disorder (ADHD) (*Cortese and Tessari, 2017*; *Griffiths et al., 2011*). Prior to the COVID-19 pandemic, prevalence of childhood overweight and childhood obesity, respectively, was 21.3% and 5.7% in Europe (*Garrido-Miguel et al., 2019*) and 20.1% and 4.3% in Norway (*Glavin et al., 2014*). The estimated prevalence in Europe of mid-childhood emotional disorders was around 4% (*Kovess-Masfety et al., 2016*; *Sadler et al., 2018*) while the global prevalence of child and adolescent ADHD was estimated at 5% (*Sayal et al., 2018*). These rates may have increased considerably in the wake of the pandemic (*Vizard et al., 2020*). In this context, there is a clear need to understand the relationship between these factors, but it is not known if child body weight causes emotional or behavioural problems.

High BMI in childhood could affect emotional symptoms through social mechanisms, for example bullying victimization (*Puhl et al., 2017*). An impact on ADHD has been proposed via sleep disturbance and neurocognitive functioning (*Vogel et al., 2015*). However, even if children with high BMI are more likely than normal weight children to experience these symptoms, associations may not be causal. Aspects of the family environment may independently affect children's BMI and their likelihood of developing emotional and behavioural symptoms, for example socioeconomic disadvantage (*Russell et al., 2016*) and parental mental health (*Hope et al., 2019a*; *Hope et al., 2019b*). Some studies have suggested that prenatal maternal obesity may confound associations of childhood BMI with emotional and behavioural symptoms (*Sanchez et al., 2018*), although the evidence is mixed (*Li et al., 2020*; *Arafat and Minică, 2018*). Reverse causality is also plausible: depressive, anxiety or ADHD symptoms

**eLife digest** Some studies show that children with obesity are more likely to receive a diagnosis of depression, anxiety, or attention-deficit hyperactivity disorder (ADHD). But this does not necessarily mean obesity causes these conditions. Depression, anxiety, or ADHD could cause obesity. A child's environment, including family income or their parents' mental health, could also affect a child's weight and mental health. Understanding the nature of these relationships could help scientists develop better interventions for both obesity and mental health conditions.

Genetic studies may help scientists better understand the role of the environment in these conditions, but it's important to consider both the child's and their parents' genetics in these analyses. This is because parents and children share not only genes, but also environmental conditions. For example, families that carry genetic variants associated with higher body weight might also have lower incomes, if parents have been affected by biases against heavier people in society and the workplace. Children in these families could have worse mental health because of effects of their parent's weight, rather than their own weight. Looking at both child and adult genetics can help disentangle these processes.

Hughes et al. show that a child's own body mass index, a ratio of weight and height, is not strongly associated with the child's mental health symptoms. They analysed genetic, weight, and health survey data from about 41,000 8-year-old children and their parents. The results suggest that a child's own BMI does not have a large effect on their anxiety symptoms. There was also no clear evidence that a child's BMI affected their symptoms of depression or ADHD.

These results contradict previous studies, which did not account for parental genetics. Hughes et al. suggest that, at least for eight-year-olds, factors linked with adult weight and which differ between families may be more critical to a child's mental health than a child's own weight. For older children and adolescents, this may not be the case, and the individual's own weight may be more important. As a result, policies designed to reduce obesity in mid-childhood are unlikely to greatly improve the mental health of children. On the other hand, policies targeting the environmental or societal factors contributing to higher body weights, bias against people with higher weights, and poor child mental health directly may be more beneficial.

could cause higher BMI, for instance via disordered eating patterns or decreased physical activity (*Blaine, 2008*; *Martins-Silva et al., 2019*). To avoid confounding and reverse causation, recent studies have applied Mendelian randomization (MR), a causal inference approach which uses genetic variants as instrumental variables for putative risk factors (*Davies et al., 2018*). Results, principally based on adult populations, are consistent with a causal influence of BMI on ADHD (*Martins-Silva et al., 2019*) and depression (*Tyrrell et al., 2019*). They are inconclusive for anxiety, reporting both positive (*Walter et al., 2015*) and negative (*Millard et al., 2019*) predicted causal effects of body weight.

However, although MR studies avoid classical confounding and reverse causation, they can be vulnerable to other sources of bias. Specifically, estimates from 'classic' MR studies – those conducted on samples of unrelated individuals – may be affected by demographic and familial factors (*Davies et al., 2019*; *Morris et al., 2020*). Bias can firstly arise from uncontrolled population stratification, where systematic differences in genotype between individuals from different ancestral clusters correlates with differences in environmental or cultural factors. This is an example of gene-environment correlation, which can lead to biased associations of genotypes and phenotypes. Secondly, indirect genetic effects may exist whereby parental genotype influences a child's phenotype via environmental pathways, termed 'dynastic effects' or 'genetic nurture' (*Kong et al., 2018*). Thirdly, assortative mating in the parents' generation, where parents are more (or less) similar to each other than would be expected by chance, can distort genotype-phenotype associations in the child's generation. Recent work has suggested that these biases may be especially pronounced for complex social and behavioural phenotypes (*Brumpton et al., 2020*; *Howe et al., 2022*). Previously reported MR estimates of the effect of BMI on emotional and behavioural problems may therefore partly reflect demographic or familial biases rather than a causal influence of BMI. To investigate this, we used a 'within-family' Mendelian randomization (within-family MR) design. This approach uses the child's, mother's, and father's genotype data as instruments for the BMI of the child, mother, and father. Within family Mendelian

randomization estimates of the effect of the child's BMI on the outcomes are robust to demographic and family-level biases. We compared within-family MR estimates with estimates from multivariable regression of the child's outcomes on the child's, mother's and father's reported BMI, and with estimates from 'classic' Mendelian randomization (classic MR), in which the child's genotype data was used to instrument the child's BMI without controlling for the parents' genotype.

## Methods

### Study population

The Norwegian Mother, Father and Child Cohort Study (MoBa) is a population-based pregnancy cohort study over 114,500 children, 95,200 mothers, and 75,200 fathers conducted by the Norwegian Institute of Public Health (*Magnus et al., 2016*). Participants were recruited from all over Norway from 1999 to 2008, with 41% of all pregnant women invited consenting to participate. The first child was born in October 1999 and the last in July 2009. The cohort now includes over 114,500 children, 95,200 mothers, and 75,200 fathers (for more details see Appendix 1: MoBa study details). As of May 2022, genotype data which had passed quality control filters was available for 76,577 children, 53,358 fathers, and 77,634 mothers. This analysis was restricted to 40,949 mother-father-child 'trios' for whom genetic data were available for all three individuals, and at least one questionnaire had been completed.

The numbers of participants excluded are shown in a STROBE flow chart in *Appendix 1—figure 1*. From all records in MoBa (N=114,030 after removing consent withdrawals), participants were excluded if the parents had not completed any of the MoBa questionnaires used in imputation models. Of the 104,915 records remaining, there were 40,949 births for which genetic data were available and had passed QC filters for mother, father, and child (for details see Appendix 1: Genotyping and imputation, and Appendix 1: Genetic quality control). Missing values in phenotypic information for these participants were estimated using multiple imputation (details in Appendix 1: Multiple imputation). Related participants were retained, but all models were clustered by genetic family ID derived using KING software (*Manichaikul et al., 2010*). This genetic family ID groups first, second, and third-degree relatives (i.e. siblings in the parental generation and their children as well as nuclear families), in this way accounting for non-independence of observations.

### Measures

Children's BMI was calculated from height and weight values reported by mothers when the children were 8 years old. Maternal pre-pregnancy BMI was calculated from height and weight reported at ~17 weeks gestation. Father's BMI was calculated from self-reported height and weight at ~17 weeks gestation. This information was missing from around 60% of fathers, and in these cases the mother's report of the father's height and weight was used instead (observed values of BMI from the two sources were correlated at 0.98). Values of height and weight more than 4 standard-deviations from the mean were treated as outliers and coded to missing.

Depressive, anxiety, and ADHD symptoms were reported by the mother when the child was 8 years old using validated measures. For depressive symptoms, the 13-item Short Mood and Feelings Questionnaire (SMFQ) was used, for anxiety symptoms the 5-item Short Screen for Child Anxiety Related Disorders (SCARED) (*Birmaher et al., 1999*) and for ADHD symptoms the Parent/Teacher Rating Scale for Disruptive Behaviour Disorders (RS-DBD) (total score and subdomain scores for inattention and hyperactivity) (*Silva et al., 2005*). Prorated summary scores were calculated for individuals with at least 80% of item-level information. Full details of all questions asked in MoBa are available at https://mobawiki.fhi.no/mobawiki/index.php/Questionnaires.

Blood samples were obtained from both parents during pregnancy and from mothers and children (umbilical cord) at birth. Details of genotyping and genetic quality control are described in Appendix 1: Genotyping and imputation and Appendix 1: Genetic quality control. Polygenic scores (PGS) for BMI were calculated using SNPs previously associated in GWAS with BMI at $p < 5.0 \times 10^{-8}$ and weighted using the individual SNP-coefficients from the GWAS. We first constructed a PGS based on the largest existing GWAS of BMI in adults (*Yengo et al., 2018*). Since genetic influences on BMI in childhood and adulthood differ (*Silventoinen et al., 2016*) we also constructed a PGS based on a GWAS of body size in childhood as recalled by adult participants of UK Biobank (*Richardson et al., 2020*). These SNPs

have been shown in external validation samples to predict BMI in childhood better than SNPs associated with adult BMI (*Richardson et al., 2020*; *Brandkvist et al., 2021*). From the full GWAS results, we excluded SNPs not available in MoBa, then used the TwoSampleMR package (*Hemani et al., 2018b*) to identify SNPs independently associated with BMI (with a clumping threshold of $r$=0.01, LD = 10,000 kb) at $p<5.0 \times 10^{-8}$. This left 954 SNPs associated with adult BMI, and 321 associated with childhood body size. Full details of SNPs included in both PGSs are provided in *Supplementary file 1a and b*. Equivalent PGSs were derived for depression and ADHD based on SNPs previously associated with these conditions at $p<5.0 \times 10^{-8}$ in GWAS (*Wray et al., 2018*; *Demontis et al., 2019*). This was not possible for anxiety, due to few known SNPs associated with these traits at $p<0.05 \times 10^{-8}$. Details of the SNPs in the depression and ADHD PGSs are provided in *Supplementary file 1c and d*.

## Statistical analysis

Among trios with genetic data, multiple imputation by chained equations was performed in STATAv16 to estimate missing phenotypic information (details in Appendix 1: Multiple imputation of phenotypes). We used non-genetic linear regression, classic MR, and within-family MR to estimate the effects of the child's BMI on the following outcomes: depressive, anxiety, and ADHD symptoms, and subdimensions of ADHD (inattention and hyperactivity). Non-genetic regression models were adjusted for child's sex, year of birth, mother's and father's BMI, and likely confounders of observational associations: mother's and father's educational qualifications, mother's and father's depressive/anxiety symptoms (using selected items from the 25-item Hopkins Checklist *Hesbacher et al., 1980*) and ADHD symptoms (from the 6-item adult ADHD self-report scale *Kessler et al., 2005*), mother's and father's smoking status during pregnancy, and maternal parity at the child's birth. For comparability, these models also included all covariates included in genetic models: genotyping centre, genotyping chip, and 20 principal components of ancestry for the child, mother, and father (for detailed information on principal components see Appendix 1: Genetic quality control). All MR models were conducted with two-stage least squares instrumental-variable regression using Stata's ivregress, with F-statistics and $R^2$ values obtained using ivreg2. Classic MR models, which do not account for parental genotype, used the child's own PGS but not those of the parents to instrument the child's BMI. Within-family MR models were multivariable MR models, in which we used PGSs for all members of a child-mother-child trio to instrument the BMI of all three individuals (model equations are provided in Appendix 1: Model equations). Classic and within-family MR models were adjusted for the child's sex and year of birth, and the genotyping centre, genotyping chip, and the first 20 principal components of ancestry for the child, mother, and father. Given skew in outcomes variables, all models used robust standard errors (Stata's vce option) and thus made no assumptions about the distribution of outcomes. We report two sets of results, in which either the adult BMI GWAS, or the childhood body size GWAS, was used to create the BMI PGS for the child, mother, and father. Z tests of difference were used to formally compare the classic MR and within-family MR estimates. To assess the extent of assortative mating in the parental generation based on phenotype data, we ran linear regression models of standardized paternal BMI, depressive symptoms, and ADHD symptoms on standardized maternal BMI, depressive symptoms, and ADHD symptoms. We then regressed paternal polygenic scores for BMI, depression, and ADHD on maternal polygenic scores for BMI, depression, and ADHD. All models investigating assortative mating adjusted for both parents' principal ancestry components and genotyping covariates. We did not examine correlations with polygenic scores for anxiety, due to few known SNPs associated with these traits at $p<0.05 \times 10^{-8}$. All statistical tests were two-tailed.

## Sensitivity analyses

To check sensitivity of results to outliers, all analyses were repeated using log-transformed versions of outcome measures (as all symptoms scales began at 0, we added 1 to scores before log-transforming). Genetic studies designed to assess causation can be biased by horizontal pleiotropy (*Davies et al., 2018*). This is when genetic variants in a polygenic score influence the outcome via pathways which do not involve the exposure. Pleiotropic effects can inflate estimated associations, or bias estimates towards the null. Methods have been developed to test for the presence of horizontal pleiotropy by comparing SNP-specific associations of exposures and outcomes, although these tests themselves rest on assumptions (*Hemani et al., 2018a*). We therefore performed additional robustness checks based on associations of individual SNPs included in the polygenic scores with BMI in the GWAS,

and associations of the same SNPs with each outcome in MoBa. It was not computationally feasible to include individual SNPs in the imputation models, so SNP-outcome associations in MoBa were calculated using unimputed SNP data with imputed outcome data. For robustness checks of classic MR models, SNP-outcome associations were adjusted for the child's sex and birth year, and the genotyping centre, genotyping chip, and ancestry principal components of the child, mother, and father. For robustness checks of within-family MR models, SNP-outcome associations were adjusted for the child's sex and birth year, mother's and father's genotype, and the genotyping centre, genotyping chip, and principal components of the child, mother, and father. We conducted inverse-variance weighted, MR-Median, MR-Mode, and MR-Egger regression in STATAv16 with the MRRobust package (*Spiller et al., 2019*). A non-zero intercept from an MR-Egger model indicates presence of horizontal pleiotropy. We repeated main analyses without using imputed data in the sample of participants who had full genetic, exposure, outcome, and covariate data. To explore nonlinearities in associations of BMI with depression, anxiety, and ADHD symptoms, we ran non-genetic models with the child's BMI divided into quintiles. Finally, MR models were run with additional adjustment for parental education. Attenuation of classic MR estimates in these models would be consistent with confounding by aspects of the family environment linked to parental education.

## Results

This analysis was restricted to 40,949 mother-father-child 'trios' for whom genetic data were available for all three individuals, and at least one questionnaire had been completed. To assess whether participants included in the analytic sample (N=40,949) differed from the rest of the MoBa sample (N=72,742), we conducted t-tests and chi-squared tests for key characteristics at birth, BMI, and outcomes using unimputed data. There were modest differences, described in Appendix 1: Comparison of analytic sample and excluded participants. BMI did not differ for mothers, fathers or children, but children in the analytic sample had slightly lower depressive symptoms (mean SMFQ = 1.81 vs 1.91), anxiety symptoms (mean SCARED = 1.04 vs 1.00) and ADHD symptoms (mean RS-DBD ADHD = 8.4 vs 8.7). Descriptive characteristics of the full MoBa sample are in *Appendix 1—table 1*.

Descriptive statistics of the analytic sample after multiple imputation is presented in *Table 1*. The mean BMI for children was 16.3 (SD = 2.0), for mothers 24.0 (SD = 4.1), and for fathers 25.9 (SD = 3.2). Corresponding descriptive characteristics from unimputed data are included in *Appendix 1—table 2*. Both polygenic scores used to instrument BMI were strong instruments, even when used in within-family models. For the adult BMI PGS, conditional first-stage F-statistics for children, mothers, and fathers were 718.7, 1338.2, and 1272.5. The conditional $R^2$ showed that the score explained 1.7%, 3.2%, and 3.0% of the variation in BMI for children, mothers, and fathers respectively. For the childhood body size PGS, conditional first-stage F-statistics were 919.8, 1071.8, and 960.2 for children, mothers, and fathers, with the scores explaining 2.2%, 2.6% and 2.3% of the variation in BMI. The correlation of the polygenic scores for adult BMI and for childhood body size was 0.38 for children, 0.36 for mothers and 0.37 for fathers.

### Associations of BMI with depressive, anxiety, and ADHD symptoms at age 8

#### Depressive symptoms (SMFQ)

In adjusted non-genetic regression models (*Figure 2*, *Appendix 1—table 3*), children's higher BMI at age 8 was associated with slightly higher depressive symptoms. Per 5 kg/m² increase in BMI, SMFQ score was 0.05 standard deviations (SD) higher (95% CI: 0.01,0.09, p=0.02). Classic MR using the adult BMI PGS suggested that for each 5 kg/m² increase in the child's BMI, the child's SMFQ score increased by 0.45 SD (95% CI: 0.26,0.64, p<0.001). Within-family MR using the adult BMI PGS also provided some evidence for an effect (beta: 0.26 SD, 95% CI: –0.01,0.52, p=0.06), but the within-family MR estimate was less precise (70% as precise as the classic MR estimate, and 15% as precise as the OLS estimates, from the ratio of standard errors). A z test for the difference (p=0.26) indicated that the within-family MR estimate was consistent with the classic MR estimate. Using the childhood body size PGS (*Figure 3*, *Appendix 1—table 4*) there was little evidence that a child's own BMI affected their depressive symptoms from either classic MR (beta: 0.08 (95% CI: –0.07,0.22, p=0.29) or within-family

**Table 1.** Descriptive characteristics of analytic sample (N=40,949)*.

| Continuous variables | mean | SD |
|---|---|---|
| Maternal age at child's birth (years) | 30.2 | 4.4 |
| Paternal age at birth (years) | 32.6 | 5.1 |
| Maternal depressive/anxiety symptoms, Hopkins Symptoms Checklist-25 (SCL-25)[†] | 1.2 | 1.9 |
| Paternal depressive/anxiety symptoms, Hopkins Symptoms Checklist-25 (SCL-25) [‡] | 1.1 | 2.1 |
| Maternal ADHD symptoms: adult ADHD self-report scale [§] | 6.7 | 3.4 |
| Paternal ADHD symptoms: adult ADHD self-report scale [¶] | 8.3 | 3.1 |
| Maternal pre-pregnancy BMI (kg/m$^2$) | 24.0 | 4.1 |
| Paternal BMI (kg/m$^2$) | 25.9 | 3.2 |
| Child's BMI at age 8 (kg/m$^2$) | 16.3 | 2.0 |
| Child depressive symptoms age 8: Short Mood and Feelings Questionnaire (SMFQ)** | 1.9 | 2.5 |
| Child anxiety symptoms age 8: Screen for Child Anxiety Related Disorders (SCARED)[††] | 1.0 | 1.2 |
| Child ADHD symptoms age 8: Parent/Teacher Rating Scale for Disruptive Behaviour Disorders (RS-DBD) [‡‡] | 8.6 | 7.2 |
| Child ADHD symptoms (inattention) age 8: Parent/Teacher Rating Scale for Disruptive Behaviour Disorders (RS-DBD) [§§] | 5.0 | 4.1 |
| Child ADHD symptoms (hyperactivity) age 8: Parent/Teacher Rating Scale for Disruptive Behaviour Disorders (RS-DBD)[§§] | 3.6 | 3.9 |

| Categorical variables | Category | % |
|---|---|---|
| Child's sex | Male | 51.1 |
| | Female | 48.9 |
| Maternal educational qualifications | 9 year elementary education | 2.0 |
| | Up to 2 years further education | 4.1 |
| | Further education: vocational | 12.2 |
| | Further education: general studies, sixth form | 14.9 |
| | Higher education: college/university, up to 4 years | 42.8 |
| | Higher education: college/university, over 4 years | 24.0 |
| Paternal educational qualifications | 9 year elementary education | 3.4 |
| | Up to 2 years further education | 5.6 |
| | Further education: vocational | 25.1 |
| | Further education: general studies, sixth form | 12.9 |
| | Higher education: college/university, up to 4 years | 28.4 |
| | Higher education: college/university, over 4 years | 24.4 |

*Table 1 continued on next page*

*Table 1 continued*

| Continuous variables | | mean | SD |
|---|---|---|---|
| | 0 | | 46.8 |
| | 1 | | 35.7 |
| | 2 | | 14.0 |
| | 3+ | | 2.7 |
| Maternal parity at child's birth | 4+ | | 0.7 |
| | Married/registered partner | | 97.4 |
| Mother's marital status at birth | single | | 2.6 |
| | never | | 51.0 |
| | Stopped before week 17 | | 42.0 |
| | Currently, sometimes | | 2.4 |
| Mother's smoking status during pregnancy | Currently, daily | | 4.5 |

*The reasons for exclusions and numbers in each case are shown in **Appendix 1—figure 1**. Missing data in BMI, outcomes and covariates was imputed using multiple imputation by chained equations. Descriptive statistics for the unimputed data are shown in Appendix 1.

[†]Based on 5 items. Possible range: 0–15.

[‡]Based on 8 items. Possible range: 0–24.

[§]Possible range: 0–24.

[¶]Possible range: 0–24.

[**]Possible range: 0-26.

[††]Possible range: 0–10.

[‡‡]Possible range: 0–54.

[§§]Possible range: 0–27.

MR (beta: 0.02 (95%CI: –0.20,0.23, p=0.88). In summary, evidence for an effect of childhood BMI on depressive symptoms was strongest using the genetic variants for adult BMI.

## Anxiety symptoms (SCARED)

In non-genetic models (**Figure 2**, **Appendix 1—table 3**), each 5 kg/m$^2$ increase in BMI was associated with a 0.07 SD lower (95% CI: –0.11,–0.03, p=0.001) SCARED score. Using the adult BMI PGS, there was little evidence for an effect from classic MR (beta: –0.06, 95% CI: –0.25,0.12, p=0.51), or within-family MR models (beta: 0.01, 95% CI: –0.25,0.29, p=0.96). Again, the within-family MR estimate was less precise than the classic MR estimate (68% as precise), or the OLS estimate (15% as precise), and the classic and within-family MR estimates were consistent (p=0.54). Using the childhood body size PGS (**Figure 3**, **Appendix 1—table 4**), MR estimates were similar (classic MR beta: –0.04, 95%: –0.18,0.11, *P*=0.62, within-family MR beta: 0.02, 95% CI: –0.18,0.22, *P*=0.83). In summary, there was little evidence from any genetic model that childhood BMI affects anxiety symptoms.

## ADHD symptoms (RS-DBD)

In non-genetic models (**Figure 2**, **Appendix 1—table 3**) children's BMI was negatively associated with ADHD symptoms after adjusting for confounders. Per 5 kg/m$^2$ increase in BMI, ADHD symptoms from the RS-DBD were 0.07 SD lower (95% CI: –0.11,–0.03, p=0.001), with similar associations observed for the inattention or hyperactivity subscales (**Figure 2**, **Appendix 1—table 3**). Using the adult BMI PGS there was evidence from both classic and within-family MR models of a positive association of BMI and ADHD. In the classic MR model ADHD symptoms were 0.35 SD higher (95% CI: 0.17,0.53, p<0.001) per 5 kg/m$^2$ increase in BMI; the within-family MR estimate, at 0.36 SD (CI: 0.09,0.63, p=0.009) was almost identical (p for difference = 0.95). A similar pattern was seen with the inattention and hyperactivity subscales (**Figure 2**, **Appendix 1—table 3**). The within-family MR estimate was again the least precise (65% as precise as the classic MR estimate, 14% as precise as the non-genetic estimate). Using the childhood body size PGS (**Figure 3**, **Appendix 1—table 4**) there was little evidence of

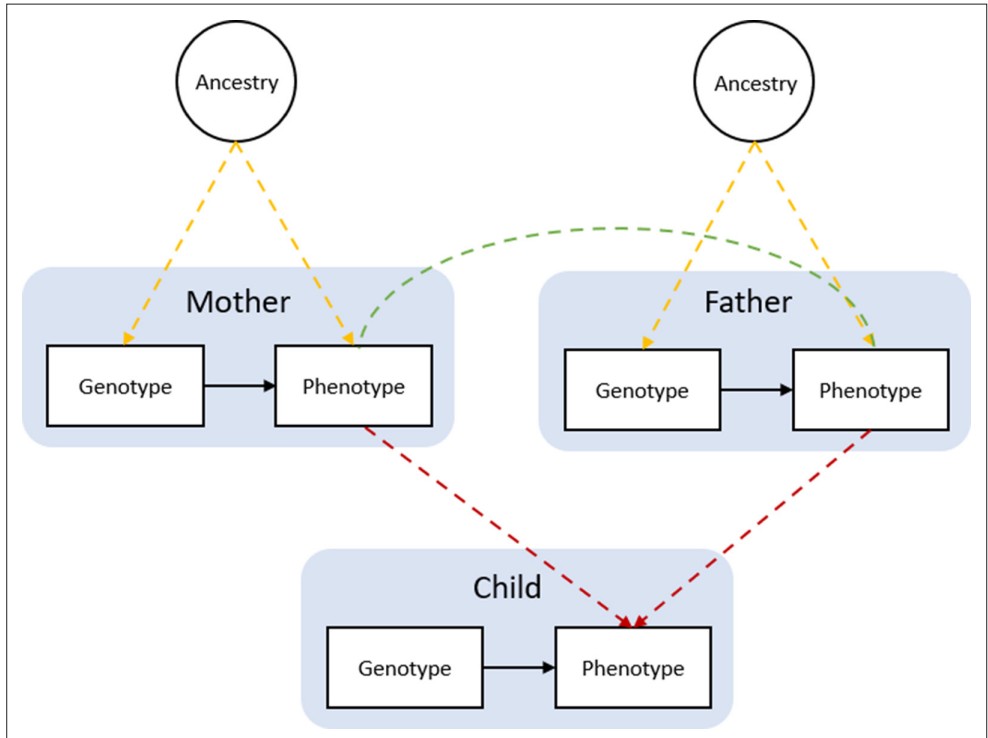

**Figure 1.** Bias in Mendelian randomization studies which do not account for parental genotype. *Figure 1* is reproduced from *Figure 1*; *Morris et al., 2020*. Population stratification due to ancestral differences (yellow lines), dynastic effects (red lines), and assortative mating (green line). In within-family Mendelian randomization, parental genotype is controlled for, so effect estimates for the influence of child's genotype on child phenotypes are unbiased by these processes.

an association from either classic MR (beta: –0.07, 95% CI: –0.21,0.07, p=0.35) or within-family MR models (beta: –0.03, 95% CI: –0.22,0.17, p=0.80). Thus, as for depressive symptoms, evidence for an effect of childhood BMI on ADHD symptoms was inconsistent and only detected using the adult BMI polygenic score.

## Association of mother's and father's BMI with child's symptoms

In non-genetic models which adjusted for the child's BMI as well as covariates, the mother's BMI was associated with slightly more depressive symptoms in the child (the child's SMFQ score was 0.05 SD higher (95% CI: 0.03,0.07, p<0.001), per 5 kg/m$^2$ increase in maternal BMI). Maternal BMI was also associated with more ADHD symptoms in the child: the child's RS-DBD score was 0.04 S.D. higher (95% CI: 0.02,0.06, p<0.001) per 5 kg/m$^2$ increase in maternal BMI, with similar associations for inattention and hyperactivity subscales. No such associations were seen with paternal BMI.

Within-family MR models also provide estimates for the effect of factors linked to maternal and paternal BMI on child outcomes, conditional on the child's own BMI. However, compared to within-family MR estimates for the child's own genotype, the interpretation of parental estimates differs. Like classic MR estimates for the child's BMI, within-family MR estimates for each parent's BMI will capture the causal effect of the parent's BMI on the child's outcome, but can also reflect residual population stratification and assortative mating in the parents' generation or earlier. For an unbiased estimate of parental effects, we would need to account for grandparental genotype. Within-family MR models provided inconsistent consistent evidence that maternal BMI affected the child's depressive symptoms: using the adult BMI PGS (*Figure 2*, *Appendix 1—table 3*), estimates suggested that higher maternal BMI increased depressive symptoms in the child (0.11 SD higher SMFQ score (95% CI: 0.02,0.19, p=0.01) per 5 kg/m$^2$ increase in maternal BMI), but within-family MR models using the childhood body size PGS did not (*Figure 3*, *Appendix 1—table 4*). There was little evidence from within-family MR of other maternal or paternal effects on the child's emotional or behavioural outcomes.

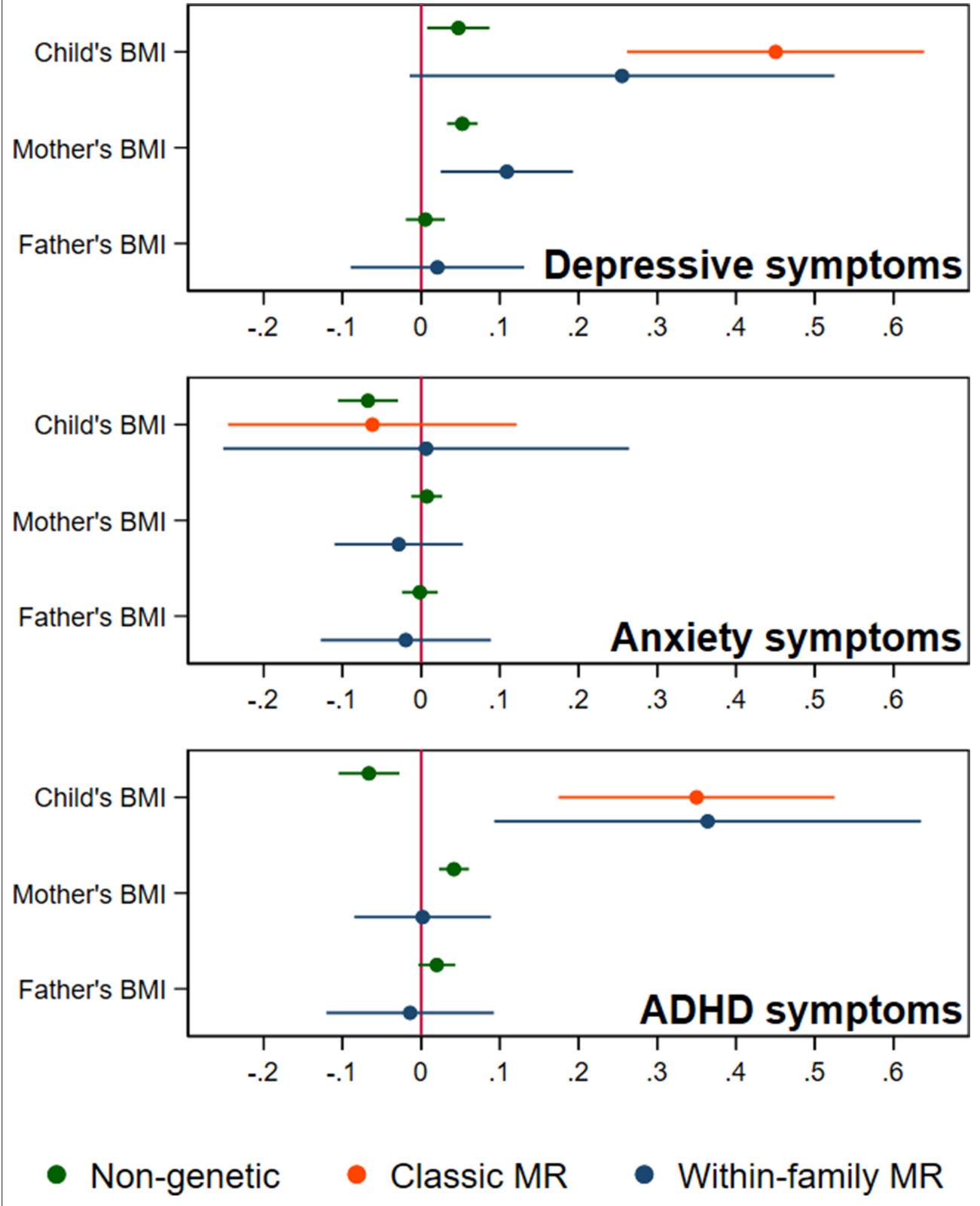

**Figure 2.** BMI and child's depressive, anxiety, and ADHD symptoms, using a polygenic score for adult BMI (N=40,949 trios). Coefficients represent standard-deviation change in outcomes per 5 kg/m² increase in BMI, shown with 95% confidence intervals.

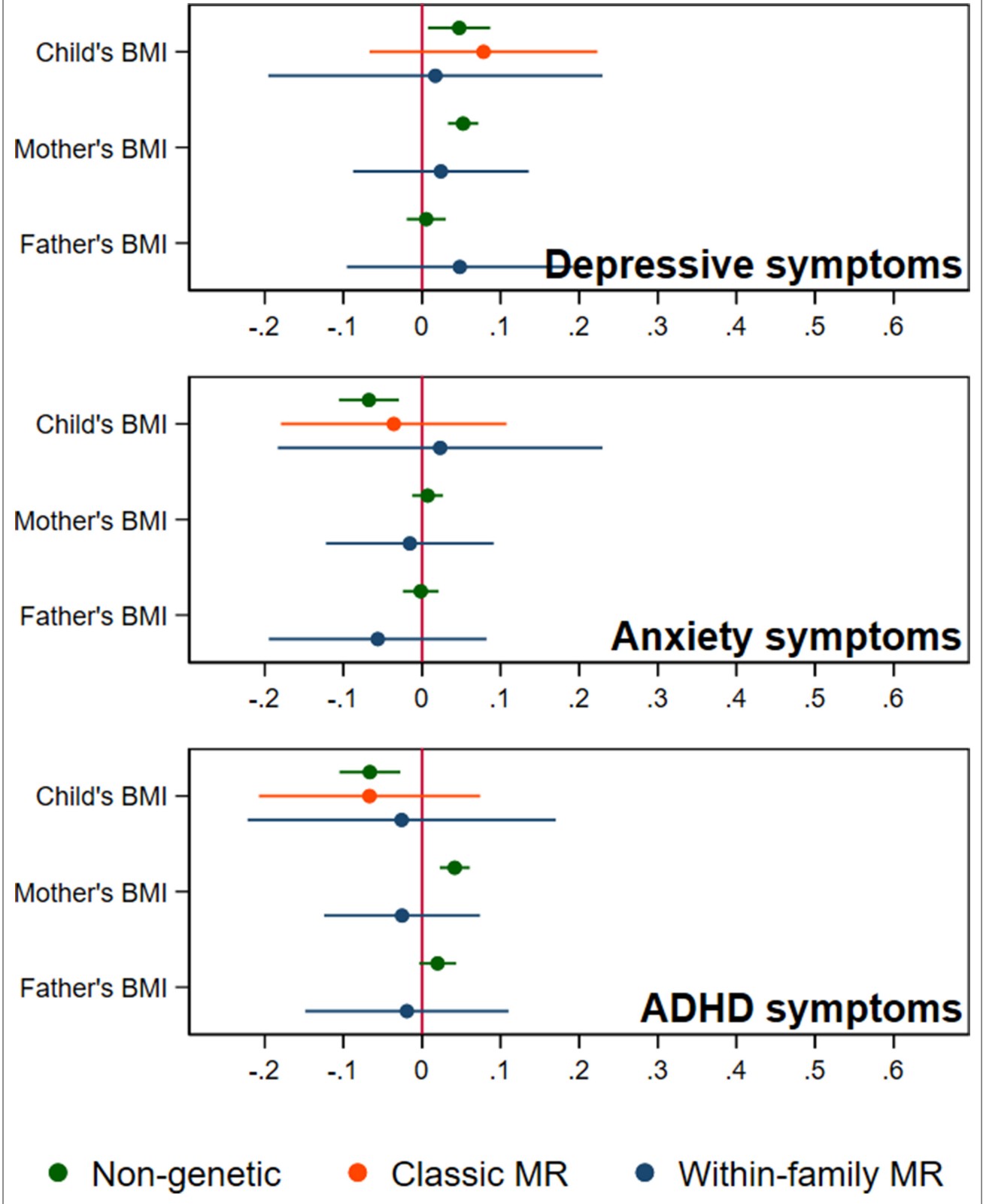

**Figure 3.** BMI and child's depressive, anxiety, and ADHD symptoms, using a polygenic score for childhood body size (N=40,949 trios). Coefficients represent standard-deviation change in outcomes per 5 kg/m² increase in BMI, shown with 95% confidence intervals.

In the parents' generation, phenotypes were associated within parental pairs, consistent with assortative mating on these traits (*Appendix 1—table 5*). Adjusted for ancestry and other genetic covariates, maternal and paternal BMI were positively associated (beta: 0.23, 95% CI: 0.22,0.25, p<0.001), as were maternal and paternal depressive symptoms (beta: 0.18, 95% CI: 0.16,0.20, p<0.001), and maternal and paternal ADHD symptoms (beta: 0.11, 95% CI: 0.09,0.13, p<0.001). Consistent with cross-trait assortative mating, there was an association of mother's BMI with father's ADHD symptoms (beta: 0.03, 95% CI: 0.02,0.05, p<0.001) and mother's ADHD symptoms with father's depressive symptoms (beta: 0.05,95% CI: 0.05,0.06, p<0.001). Phenotypic associations can reflect the influence of one partner on another as well as selection into partnerships, but regression models of paternal polygenic scores on maternal polygenic scores also pointed to a degree of assortative mating. Adjusted for ancestry and genotyping covariates, there were small associations between parents' BMI polygenic scores (beta: 0.01, 95% CI: 0.00,0.02, p=0.02 for the adult BMI PGS, and beta: 0.01, 95% CI: 0.00,0.02, p=0.008 for the childhood body size PGS), and of the mother's childhood body size PGS with the father's ADHD PGS (beta: 0.01, 95% CI: 0.00,0.02, p=0.03). We did not detect associations with pairs of other polygenic scores, which may be due to insufficient statistical power.

## Sensitivity analyses

Analyses using log-transformed versions of the outcomes (*Appendix 1—Tables 6 and 7*) were consistent with main results. Robustness checks based on comparing associations of individual SNPs with BMI in the GWAS and with children's outcomes in MoBa (*Appendix 1—Tables 8 and 9*) were consistent with the main results. MR-Egger models found little evidence of horizontal pleiotropy, although MR-Egger estimates were imprecise (*Appendix 1—Tables 8 and 9*). Results of analyses using the complete-case sample were qualitatively similar to results using imputed data (*Appendix 1—Tables 10 and 11*). In non-genetic models where the child's BMI was divided into quintiles (*Appendix 1—table 12*), there was little evidence of nonlinear associations. With additional adjustment for parental education, point estimates for depressive and ADHD symptoms in classic MR models were closer to the null, but confidence intervals substantially overlapped (*Appendix 1—Tables 13 and 14*).

# Discussion

In a large cohort of Norwegian 8-year-olds, higher childhood BMI was phenotypically associated with slightly more depressive symptoms, but fewer anxiety symptoms and ADHD symptoms. Genetic analyses using the adult BMI PGS suggested that higher BMI in childhood increased symptoms of both depression and ADHD. This was clearest in classic MR models, but also suggested by within-family MR models, whose precision is lower but which account for parental genotype. Compared to associations from non-genetic models, effect sizes for depression and ADHD from genetic models based on the adult BMI PGS were larger. However, these estimates were less precise, and confidence intervals for the classic MR and within-family MR estimates substantially overlapped for all outcomes. The childhood body size PGS explained more variation in children's BMI than the adult BMI PGS did, consistent with other studies (*Richardson et al., 2020*; *Brandkvist et al., 2021*), while the adult BMI PGS explained more variation in maternal and paternal BMI. Genetic analyses which used the childhood body size SNPs provided little evidence that the child's BMI affected their depressive or ADHD symptoms outcomes. This suggests that genetic variation associated with adult BMI has a greater impact on these outcomes than genetic variation associated with recalled childhood body size. This is consistent with the moderate correlation observed between the two polygenic scores, indicating that they capture both overlapping and unique variation. Our results may therefore reflect differences in how each set of SNPs relate to traits other than childhood BMI which are relevant to a child's depressive and ADHD symptoms. Nevertheless, within-family MR estimates using the childhood body size PGS were still consistent with small effects of the child's BMI on all outcomes, with upper confidence limits around a 0.2 standard-deviation increase in each outcome per 5 kg/m² increase in BMI. There was little evidence that maternal or paternal BMI affected a child's ADHD or anxiety symptoms. In within-family MR models using the adult BMI PGS, but not the childhood body size PGS, maternal BMI was positively associated with children's depressive symptoms. This is consistent with a causal impact of the mother's recent BMI but not their BMI in childhood, but it may also reflect family-level biases from previous generations.

The positive association between BMI and depressive symptoms in non-genetic models accords with previous observational studies (*Lindberg et al., 2020*; *Patalay and Hardman, 2019*; *Quek et al., 2017*; *Geoffroy et al., 2014*). The inverse association between BMI and anxiety symptoms in non-genetic models contrasts with the results of a recent study, in which Swedish 6–17 year olds receiving treatment for obesity had a greater likelihood of a diagnosis or prescription for anxiety disorder compared to controls (*Lindberg et al., 2020*). The discrepancy may reflect confounding (we adjusted for more factors, including parental BMI), age of the participants (children in our study were younger) or differences in the outcome or exposure, since we considered anxiety symptoms rather than diagnosis, and a continuous BMI measure rather than obesity. However, anxiety symptoms in our sample were not raised in the top BMI quintile. Another difference concerns the population: children receiving obesity treatment may be more likely than other children with obesity to experience anxiety symptoms or to receive a diagnosis. The inverse association between BMI and ADHD symptoms in non-genetic models contrasts with previous reports of positive or null associations with obesity, which typically adjusted for fewer confounders (*Cortese and Tessari, 2017*; *Nigg et al., 2016*). Since previous studies have found more evidence of an association in adults than children, and often considered ADHD diagnoses rather than symptoms, the discrepancy may also point to age-varying associations, or to different influences on likelihood of diagnosis compared to parent-reported symptoms (*Nigg et al., 2016*; *Cortese and Tessari, 2017*).

For depressive symptoms and ADHD, classic and within-family MR estimates using the adult BMI PGS were larger than estimates from non-genetic models. Horizontal pleiotropy, which we could not rule out, could have inflated MR estimates. It could also help explain the discrepancy in results using the adult BMI and childhood body size polygenic scores, if SNPs in the adult BMI polygenic score have a greater impact on depressive or ADHD symptoms via pathways independent of childhood BMI. We found little evidence of pleiotropy using MR-Egger estimators, but the power to detect pleiotropy with this method is low. Additionally, classic MR estimates may be inflated by demographic and familial factors, but within-family MR estimates for effects of a child's own BMI are robust to these factors. For depressive symptoms, the within-family MR estimate was closer to the non-genetic estimate than the classic MR estimate, which may reflect bias in the classic MR estimate due to demographic and familial factors. At the same time, the within-family MR estimate was imprecise, and confidence limits consistent with a substantial effect of children's BMI on depressive symptoms. For ADHD, point estimates from the classic MR and within-family MR models using the adult BMI PGS were very similar, and both statistically distinguishable from the null. These results therefore accord with a recent study which accounted for family-level biases by using dizygotic twin pairs, obtaining between-family and within-family estimates for the effect of BMI on ADHD symptoms (*Liu et al., 2021*). Using a PGS of SNPs associated with adult BMI, within-family analysis found a 0.07 S.D. increase in ADHD symptoms at age 8 per S.D. increase in BMI PGS, which was consistent with the between-family estimate. The between-family estimate was attenuated by adjustment for parental education, suggesting an influence of family-level processes. In our study, classic MR estimates for depressive and ADHD symptoms which adjusted for parental education were consistent with the main results, with largely overlapping confidence intervals, although point estimates were closer to the null. Thus, our results are also consistent with an influence of demographic or family-level effects, and with earlier evidence that such processes impact the relationship between BMI and ADHD (*Chen et al., 2014*; *Geuijen et al., 2019*).

Several sources of genetic familial bias may have influenced classic MR estimates of the impact of the child's own BMI. Firstly, frequencies of BMI-associated variants may differ between sub-populations in a similar manner to environmental influences on emotional or behavioural functioning (population stratification). Such gene-environment correlation can inflate estimates from classic MR models, but are unlikely to affect within-family MR models, where ancestry is fully controlled for via parental genotypes. Although we included principal components of ancestry in all models, residual population stratification may nevertheless have influenced the classic MR results. Secondly, there may be indirect effects of parental BMI via the family environment (dynastic effects, or genetic nurture). This could explain the association of maternal BMI with children's depressive symptoms in the within-family MR model using the adult BMI PGS. In observational studies, maternal pre-pregnancy obesity is linked with children's risk of emotional disorders and ADHD (*Sanchez et al., 2018*). Although mechanisms are not well understood, an in utero effect on children's neurodevelopment of metabolic correlates of obesity has been proposed (*Edlow, 2017*). Our within-family MR results suggest that previously

reported associations of maternal BMI with a child's ADHD are not causal, but are consistent with an effect on the child's depressive symptoms. This could reflect an impact of maternal BMI later in the child's life. A well-documented 'wage penalty' exists for high BMI (*Howe et al., 2020*), especially for women (*Bozoyan and Wolbring, 2018*) reflecting social consequences of obesity being a stigmatized condition (*Giel et al., 2010*). High BMI in adulthood is also linked to worse mental health, with stronger associations for women again pointing to gendered social processes (*Rubino et al., 2020*). Maternal BMI may therefore influence children's emotional and behavioural problems via economic consequences, or via maternal mental health, throughout childhood. However, while our results are consistent with an influence of maternal BMI on child's depressive symptoms, these results should be interpreted with caution. In contrast to estimated effects for the child's BMI, where controlling for parental genotype is likely to eliminate familial biases, estimated maternal and paternal effects from within-family MR models may have been impacted by familial biases in previous generations. Adjustment for grandparental genotype would be required to obtain similarly unbiased estimates for the parents. Thirdly, people with high BMI may be more likely to partner with people with emotional or behavioural conditions (cross-trait assortative mating). Over generations, this would induce an association of not only the phenotypes but of associated genetic variants. We found some genomic evidence of assortative mating for BMI, and cross-trait assortative mating between BMI and ADHD, but not between other traits. However, associations between polygenic scores, which only capture some of the genetic variation associated with these phenotypes, may not capture the full extent of genetic assortment on these traits.

Despite a high participation rate, MoBa is not perfectly representative, and selection biases linked to participation could have affected our results. The current analyses were restricted to families with complete genetic data and at least some relevant questionnaire data. These families were found to have slightly more years of education than the wider MoBa sample, and the children to score slightly lower for depressive, anxiety, and ADHD symptoms. Reflecting the requirement of genetic data for fathers, single mothers were under-represented. Analyses were restricted to individuals of European ancestry, with polygenic scores based on results of GWAS which were also restricted to individuals of European ancestry. Consequently, our results may not be generalisable to other populations. Outcomes were based on mother-reported symptoms of depression, anxiety disorders and ADHD, and estimates based on diagnoses may have differed. However, a child's sociodemographic characteristics can influence their likelihood of diagnosis independently of symptoms (*Thompson et al., 2021*), indicating that such an approach is not always preferable. BMI measurements were based on reported height and weight, so reporting bias may have influenced relationships. In many families, fathers' BMI was based on height and weight reported by the mothers. However, these measures were very highly correlated with father's self-reports, so additional measurement error is unlikely to have greatly affected our results for father's BMI. Due to attrition, a substantial proportion of values for the child's BMI and outcomes were imputed, and we cannot be sure that observations were missing at random conditional on variables included in imputation models. Effects of parental BMI may be time-varying, for example a parent's own BMI during childhood could influence their child independent of the parent's later BMI. We could not explore these effects because information on parent's childhood BMI was not available. Within-family MR may still be affected by horizontal pleiotropy, and recent genetic work points to genetic overlap between BMI and psychiatric disorders including major depression (*Bahrami et al., 2020*). While robustness checks found little evidence of pleiotropy, these methods rely on assumptions. Moreover, MR-Egger is known to give imprecise estimates (*Burgess and Thompson, 2017*), and confidence intervals from MR-Egger models were wide. Thus, pleiotropy cannot be ruled out. The Mendelian randomization methods employed here assume any causal impact of BMI is linear – that a kg/m² increase in BMI will have the same impact regardless of the child's initial BMI. There is substantial evidence for a 'J-shaped' phenotypic association of BMI with common mental disorders, consistent with an impact of both high and low BMI on risk of depression or anxiety (*McCrea et al., 2012*; *Gaysina et al., 2011*; *Geoffroy et al., 2014*). Genetic methods exist for exposures with nonlinear effects but require much larger samples (*Sun et al., 2019*). If there exist nonlinear effects of BMI on mental health, rather than vice versa, our results may underestimate the effects of high BMI. Finally, the effects of BMI on emotional and behavioural functioning likely differ by age, and relationships may be substantially different for older children or adolescents. In particular, depressive symptoms do not tend to occur until the teenage years (*Kwong et al., 2019*)

and observational associations of BMI and ADHD become clearer with age (*Nigg et al., 2016*). Work in larger samples of related individuals will be needed to precisely estimate the influence of a child's BMI on their emotional and behavioural outcomes. In response to a reviewer's request, we conducted post-hoc power calculations to estimate the minimum effect on the outcomes of the child's BMI which could be detected with 80% power in a dataset of this size (40,949 trios). Simulations indicated that, using the adult BMI PGS, an effect on each outcome of 0.15 S.D. per 5 $kg/m^2$ could be detected using classic MR, and an effect on each outcome of 0.22 S.D. per 5 $kg/m^2$ using within-family MR. Using the childhood body size PGS, equivalent detectable effects were 0.12 S.D. and 0.16 S.D. per 5 $kg/m^2$. MoBa is currently the largest individual study in which this approach can be applied, but new data are becoming available which will allow analyses of this kind within and across studies, such as through the Within Family Consortium https://www.withinfamilyconsortium.com/home/. Meanwhile, studies with extensive intergenerational information will be needed to fully explore mechanisms linking child outcomes to maternal BMI.

## Conclusion

Our results suggest that genetic variation associated with BMI in adulthood affects a child's depressive and ADHD symptoms, but genetic variation associated with recalled childhood body size does not substantially affect these outcomes. There was little evidence that BMI affects anxiety. However, our estimates were imprecise, and these differences may be due to estimation error. There was little evidence that parental BMI affects a child's ADHD or anxiety symptoms, but factors associated with maternal BMI may independently influence a child's depressive symptoms. Genetic studies using unrelated individuals, or polygenic scores for adult BMI, may have overestimated the causal effects of a child's own BMI.

## Acknowledgements

The Norwegian Mother, Father and Child Cohort Study is supported by the Norwegian Ministry of Health and Care Services and the Ministry of Education and Research. We are grateful to all the participating families in Norway who take part in this on-going cohort study. We thank the Norwegian Institute of Public Health (NIPH) for generating high-quality genomic data. This work was performed on the TSD (Tjeneste for Sensitive Data) facilities, owned by the University of Oslo, operated and developed by the TSD service group at the University of Oslo, IT-Department (USIT). (tsd-drift@usit.uio.no). The analyses were performed on resources provided by Sigma2 - the National Infrastructure for High Performance Computing and Data Storage in Norway. This research is part of the HARVEST collaboration, supported by the Research Council of Norway (#229624). We also thank the NORMENT Centre for providing genotype data, funded by the Research Council of Norway (#223273) and deCODE Genetics, South East Norway Health Authority and KG Jebsen Stiftelsen. We further thank the Center for Diabetes Research, the University of Bergen for providing genotype data and performing quality control and imputation of the data funded by the ERC AdG project SELECTionPREDISPOSED, Stiftelsen Kristian Gerhard Jebsen, Trond Mohn Foundation, the Research Council of Norway, the Novo Nordisk Foundation, the University of Bergen, and the Western Norway Health Authorities (Helse Vest). Funding: This research was funded by a project entitled 'social and economic consequences of health: causal inference methods and longitudinal, intergenerational data', which is part of the Health Foundation's Social and Economic Value of Health Programme (Grant ID: 807293). The Health Foundation is an independent charity committed to bringing about better health and health care for people in the UK. This research is part of the HARVEST collaboration, supported by the Research Council of Norway (#229624). Individual co-authors area also supported by specific sources of funding. ZA is supported by a Marie Skłodowska-Curie Fellowship from the European Union (894675) and the South-Eastern Norway Regional Health Authority (2019097). TR is supported by the Research Council of Norway (274611 PI: Reichborn-Kjennerud). OAA is funded by the Research Council of Norway (223273) and EU H2020 RIA (847776 CoMorMent). ØH is supported by the University of Bergen, Norway. SJ was supported by Helse Vest's Open Research Grant (grants #912250 and F-12144), the Novo Nordisk Foundation (grant NNF19OC0057445) and the Research Council of Norway (grant #315599). PN is supported by the European Research Council (AdG SELECTionPREDISPOSED #293574), the Trond Mohn Foundation (Mohn Center for Diabetes Precision Medicine), the Research Council of Norway (FRIPRO grant #240413), the Western Norway Regional Health Authority (Strategic Fund "Personalized Medicine for Children and Adults"), the Novo Nordisk Foundation (grant #54741), and the

Norwegian Diabetes Association. AH was supported by the South-Eastern Norway Regional Health Authority (2018059, 2020022) and the Research Council of Norway (288083). LDH is supported by a Career Development Award from the UK Medical Research Council (MR/M020894/1). NMD was supported via a Research Council of Norway grant (295989). The Medical Research Council (MRC) and the University of Bristol support the MRC Integrative Epidemiology Unit [MC_UU_00011/1] (AMH, ES, LDH, NMD, GDS, TM). The funders had no role in the design or execution of this analysis, interpretation of results, or the decision to publish.

## Additional information

### Competing interests

Ole A Andreassen: has received speaker's honorarium from Sunovion and Lundbeck and is a consultant for HealthLytix. The other authors declare that no competing interests exist.

### Funding

| Funder | Grant reference number | Author |
|---|---|---|
| Medical Research Council | MC_UU_00011/1 | Amanda M Hughes<br>Eleanor Sanderson<br>Tim Morris<br>George Davey Smith<br>Laura D Howe<br>Neil M Davies |
| University of Bristol | | Amanda M Hughes<br>Eleanor Sanderson<br>Tim Morris<br>George Davey Smith<br>Laura D Howe<br>Neil M Davies |
| European Research Council | 894675 | Ziada Ayorech |
| South-Eastern Norway Regional Health Authority | 2019097 | Ziada Ayorech |
| Research Council of Norway | 295989 | Neil M Davies |
| Research Council of Norway | 288083 | Alexandra Havdahl |
| South-Eastern Norway Regional Health Authority | 2018059 | Alexandra Havdahl |
| Research Council of Norway | 274611 | Ted Reichborn-Kjennerud |
| Research Council of Norway | 223273 | Ole A Andreassen |
| European Research Council | 847776 CoMorMent | Ole A Andreassen |
| University of Bergen | | Øyvind Helgeland |
| South-Eastern Norway Regional Health Authority | 2020022 | Alexandra Havdahl |
| Helse Vest | 912250 | Stefan Johansson |
| Helse Vest | F-12144 | Stefan Johansson |
| Novo Nordisk | NNF19OC0057445 | Stefan Johansson |
| Research Council of Norway | 315599 | Stefan Johansson |
| Medical Research Council | MR/M020894/1 | Laura D Howe |
| European Research Council | 293574 | Pål Njølstad |
| Trond Mohn stiftelse | | Pål Njølstad |
| Research Council of Norway | 240413 | Pål Njølstad |

| Funder | Grant reference number | Author |
|---|---|---|
| Western Norway Regional Health Authority | | Pål Njølstad |
| Novo Nordisk | 54741 | Pål Njølstad |
| Norwegian Diabetes Association | | Pål Njølstad |

The funders had no role in study design, data collection and interpretation, or the decision to submit the work for publication.

## Author contributions

Amanda M Hughes, Conceptualization, Formal analysis, Investigation, Visualization, Writing – original draft, Writing – review and editing; Eleanor Sanderson, Formal analysis, Methodology, Writing – review and editing; Tim Morris, Ziada Ayorech, Martin Tesli, Helga Ask, Øyvind Helgeland, Stefan Johansson, George Davey Smith, Investigation, Writing – review and editing; Ted Reichborn-Kjennerud, Data curation, Investigation, Writing – review and editing, Funding acquisition; Ole A Andreassen, Per Magnus, Pål Njølstad, Data curation, Investigation, Writing – review and editing; Alexandra Havdahl, Conceptualization, Data curation, Funding acquisition, Investigation, Writing – review and editing; Laura D Howe, Funding acquisition, Writing – review and editing, Conceptualization; Neil M Davies, Conceptualization, Funding acquisition, Investigation, Writing – review and editing

## Author ORCIDs

Amanda M Hughes http://orcid.org/0000-0001-5896-7650
Tim Morris http://orcid.org/0000-0001-8178-6815
Helga Ask http://orcid.org/0000-0003-0149-5319
George Davey Smith http://orcid.org/0000-0002-1407-8314
Neil M Davies http://orcid.org/0000-0002-2460-0508

## Ethics

The establishment of MoBa and initial data collection was based on a license from the Norwegian Data Protection Agency and The Regional Committees (REC) for Medical and Health Research Ethics. The REC South East Norway, one of four in Norway, was the ethical committee that evaluated the ethics of this study. Approval from the REC was granted (2016/1702). Informed consent was obtained from each MoBa participant upon recruitment, which included consent to link to the Medical Birth Registry of Norway (MBRN). The MoBa cohort is now based on regulations related to the Norwegian Health Registry Act.

## Decision letter and Author response

Decision letter https://doi.org/10.7554/eLife.74320.sa1
Author response https://doi.org/10.7554/eLife.74320.sa2

# Additional files

## Supplementary files

• Supplementary file 1. (a) SNPs used in the polygenic score for adult BMI. (b) SNPs used in the polygenic score for childhood body size BMI. (c) SNPs used in the polygenic score for depression. (d) SNPs used in the polygenic score for ADHD.

• Transparent reporting form

• Reporting standard 1. Strobe checklist.

## Data availability

The consent given by the participants does not open for storage of data on an individual level in repositories or journals. Researchers who want access to data sets for replication should submit an application to datatilgang@fhi.no. Access to data sets requires approval from The Regional Committee for Medical and Health Research Ethics in Norway and an agreement with MoBa.

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

# Appendix 1

## Additional methods

### 1. MoBa study details

This study is based on the Norwegian Mother, Father and Child Cohort Study (MoBa) and uses data from the Medical Birth Registry of Norway (MBRN). The Medical Birth Registry (MBRN) is a national health registry containing information about all births in Norway. The current analysis is based on version 12 of the quality-assured data files released for research in January 2019. The establishment of MoBa and initial data collection was based on a license from the Norwegian Data Protection Agency and approval from The Regional Committees for Medical and Health Research Ethics. The MoBa cohort is now based on regulations related to the Norwegian Health Registry Act. The current study was approved by The Regional Committees for Medical and Health Research Ethics (2016/1702).

The Norwegian Mother, Father and Child Cohort Study is supported by the Norwegian Ministry of Health and Care Services and the Ministry of Education and Research. We are grateful to all the participating families in Norway who take part in this on-going cohort study.

### 2. Genotyping

Genotyping of MoBa has been conducted through multiple research projects, spanning several years. The research projects (HARVEST, SELECTIONpreDISPOSED, and NORMENT) provided genotype data to MoBa Genetics. In total, 238,001 MoBa samples were sent to be genotyped in 24 genotyping batches. This was carried out at 3 centres (1. Genomics Core Facility, Trondheim, Norway, 2. ERASMUS MC, Rotterdam, Netherlands, and 3. deCODE Genetics, Reykjavik, Iceland) using six genotyping arrays. The 24 batches had varying selection criteria; this included a batch of ADHD child cases and their parents, and another of matched control children and their parents. Detailed information on batch selection and the genotyping process are described elsewhere (*Corfield et al., 2022*).

### 3. Genetic quality control

Pre-imputation QC, phasing, imputation, and post-imputation QC were carried out according to the MoBaPsychGen pipeline, which includes QC on both single nucleotide polymorphism (SNP) and individual level, and whose full details are provided elsewhere (*Corfield et al., 2022*). Output of the pipeline included 207,569 unique individuals and 6,981,748 SNPs after post-imputation QC; unique individuals comprised 76,577 children, 53,358 fathers, and 77,634 mothers. Phasing and imputation were performed using the publicly available Haplotype Reference Consortium release 1.1 panel as a reference. Information from the Medical Birth Registry of Norway and MoBa questionnaires were used to identify biological sex, year of birth, reported parent-offspring (PO) relationships, and, in the offspring generation, multiple births. Ancestry outliers were identified based on principal component analysis with the 1000 Genomes phase 1 unrelated data (1,083 individuals) (*Auton et al., 2015*). Approximately 95% of the participants were identified as having European ancestry. Ancestry outliers were removed based on visual inspection, using pairwise plots for the first seven principal components. Similarly, principal component analysis was used to identify substructure within each subpopulation of all MoBa batches. PCs were first estimated in founders only, and non-founders projected into the PC space of the founders. Outliers were removed based on visual inspection, using pairwise plots for the first ten principal components.

Relatedness was inferred using KING version 2.2.5 (*Manichaikul et al., 2010*). In the presence of admixture, KING accurately infers MZ twin or duplicate pairs (kinship coefficient >0.3540), first-degree (PO, FS, DZ twin pairs; kinship coefficient range 0.1770–0.3540), second-degree (HS, GO, AUNN; kinship coefficient range 0.0884–0.1770), and third-degree (first cousins; kinship coefficient range 0.0442–0.0884) relationships. After within-family and between-family relationships were confirmed by genetic data, a check for Mendelian errors (ME) was performed in PLINK. The ME check included families with one or two parents present in the data. Families with >5% errors and SNPs with >1% errors were removed. The remaining ME were set to missing.

### 4. Multiple imputation

Multiple imputation by chained equations was performed in STATAv16 to estimate missing phenotypic information for the 40,949 trios with complete genetic data. 100 imputed datasets were

produced and analysis across these datasets conducted with STATA's mi estimate commands. The imputation model included all BMI variables used in the main analyses (child's BMI at age 8, mother's pre-pregnancy BMI and father's BMI as reported at 17 weeks gestation), the child's sex and year of birth, and other phenotypic covariates used in non-genetic models, including mother's and father's smoking status reported at 17 weeks gestation, mother's and father's depressive/anxiety symptoms [using selected items from the 25-item Hopkins Checklist (*Hesbacher et al., 1980*)], and ADHD symptoms [from the 6-item adult ADHD self-report scale (*Kessler et al., 2005*)], maternal parity at the child's birth, and family socioeconomic characteristics, including parental educational qualifications and categorical variables of income and subjective financial strain. Variables from the birth registry file were also included as auxiliary variables: the mother's marital status, the age of the mother and father, and the child's birthweight and length. Approximately normally-distributed continuous variables including BMI were imputed using truncated regression, specifying as upper and lower limits the smallest and largest values observed in the full MoBa sample. Ordered categorical variables were imputed with ordered logistic regression. There was no missingness in genetic information within the analytic sample. Polygenic scores for adult BMI, childhood body size, depression, ADHD, and educational attainment were included on the right-hand side of the imputation equations, along with indicators for genotyping centre and chip and the 20 principal components of ancestry for all individuals. Continuous variables which were not normally-distributed were imputed with predictive mean matching, specifying knn(10). This included child's depressive and anxiety symptoms at age 8 (SMFQ and SCARED summary scores), mother's and father's depressive/anxiety symptoms at 17 weeks gestation (summary scores based on items from the 25-item Hopkins Checklist (*Hesbacher et al., 1980*), and mother's and father's ADHD symptoms from the 6-item adult ADHD self-report scale (*Kessler et al., 2005*). To facilitate analysis of ADHD inattention and hyperactivity subscales, the two subscales were imputed, again with predictive mean matching, and the full scale calculated post-imputation with mi passive. An earlier measure of the child's ADHD symptoms at 5 yrs, based on questions from the Short-Form Conners Parent Rating Scale (*Kumar and Steer, 2003*), was included as an auxiliary variable. The percentage of imputed data in the analytic sample for each variable was: mother's education 3.4%, father's education 2.2%, mother's smoking 2.0%, father's smoking 0.7%, mother's depressive symptoms 3.2%, father's depressive symptoms 7.1%, mother's ADHD symptoms 40.9%, father's ADHD symptoms 60.1%, mother's BMI: 4.0%, father's BMI: 3.8%, child's BMI at age 8: 60.6%, child's depressive symptoms at age 8: 54.2%, child's anxiety symptoms at age 8: 54.1%, child's ADHD symptoms (inattention and hyperactivity) both 54.1%.

## 5. Polygenic score construction

Beginning with the full GWAS results for each phenotype (*Yengo et al., 2018*; *Richardson et al., 2020*; *Wray et al., 2018*; *Demontis et al., 2019*), R was used to subset SNPs included in full GWAS results to SNPs also available in the quality controlled MoBa data. From SNPs available in both, independent genome-wide significant associations were identified by clumping in MRBase, specifying $r=0.01$ and $p<5.0 \times 10^{-8}$ (*Hemani et al., 2018b*). First subsetting to SNPs available in MoBa and then clumping within these avoids the need for an additional step identifying proxy SNPs. This left 954 SNPs associated with adult BMI, and 321 associated with childhood body size.

Dosage data for MoBa participants for the genetic variants relevant to each polygenic score were extracted from the final quality controlled -bfiles using PLINK, and .raw files imported to STATA. SNPs were harmonized by comparing effect alleles in the GWAS and reference alleles in MoBa.

Polygenic scores were calculated as the sum of the number of effect alleles (0, 1, or 2) multiplied in each case by the harmonized SNP-coefficient from the GWAS. There was a very small amount of missingness in individual SNPs due to quality control measures. Among SNPs included in the adult BMI PGS, the median proportion of individuals missing the SNP was 0.001 and the maximum 0.04. For SNPs included in the childhood body size PGS, equivalent values were 0.002 and 0.04, for SNPs included in the depression and the ADHD PGS these were 0.003 and 0.01. Where individual SNP information was missing for an individual, the sample mean for the number of effect alleles (between 0 and 2) was added for that SNP.

## 6. Model equations

In within-family MR models, we used polygenic scores for all members of a child-mother-child trio to instrument the BMI of all three individuals. Within-families MR models were adjusted for child's sex, the first 20 principal components of ancestry for the child and both parents, and the genotyping chip and centre of the child and both parents.

In Stata, this was specified in the form: ivregress 2sls outcome (c_bmi m_bmi f_bmi = c_pgs m_pgs f_pgs) sex c_PC* m_PC* f_PC* c_genotyping_centre* m_ genotyping_centre* f_ genotyping_centre* c_genotyping_chip* m_ genotyping_chip* f_ genotyping_chip*

This can be represented with the following equations:

The outcomes (depressive, anxiety, and ADHD symptoms) can be expressed as:

$y_i = \beta_0 + \beta_1 X_{ic} + \beta_2 X_{im} + \beta_3 X_{if} + \beta_C C + e_i$

The three exposures are offspring's, mother's, and father's BMI:

$X_{ic} = \gamma_{oc} + \gamma_{1c} Z_{ic} + \gamma_{2c} Z_{im} + \gamma_{3c} Z_{if} + \gamma_{4c} C + v_{ic}$

$X_{im} = \gamma_{om} + \gamma_{1m} Z_{ic} + \gamma_{2m} Z_{im} + \gamma_{3m} Z_{if} + \gamma_{4m} C + v_{im}$

$X_{if} = \gamma_{oc} + \gamma_{1f} Z_{ic} + \gamma_{2f} Z_{im} + \gamma_{3f} Z_{if} + \gamma_{4f} C + v_{if}$ where: $y_i$ = outcome

$X_{ic}$ = child's BMI

$X_{im}$ = mother's BMI

$X_{if}$ = father's BMI

$Z_{ic}$ = child's polygenic score

$Z_{im}$ = mother's polygenic score

$Z_{if}$ = father's polygenic score

C=covariates including principal components for offspring, mother and father and genotyping centre and chip $e_i$ = error term for the outcome equation $v_{ic}$ = error term for the child exposure (BMI) equation $v_{im}$ = error term for the mother exposure (BMI) equation $v_{if}$ = error term for the father exposure (BMI) equation

## Additional Results

### 1. Comparison of analytic sample and excluded participants

To assess if participants included in the analytic sample (N=40,949) differed from others in the birth registry file (N=72,742) (*Appendix 1—figure 1*), we conducted t-tests and chi-squared tests for key characteristics at birth, BMI, and outcomes using unimputed data. Reflecting the large size of the sample population, several differences reached statistical significance. Children in the analytic sample did not differ from those excluded on sex but were slightly older (mean year of birth: 2005.4 vs 2004.5). They had slightly higher birthweight (mean = 3.6 kg vs 3.4 kg) and slightly younger fathers (mean paternal 32.6 vs 32.8 years). Mothers and fathers included in the analytic sample had slightly higher educational qualifications (e.g., 24.1% and 24.3% of mothers and fathers respectively had 4 years of college education, against 21.0% and 21.5% for those not included). At the time of the child's birth, mothers in the analytic sample had fewer existing children (e.g., 46.8% vs 42.4% had none), were more likely to be married or cohabiting (97.4% vs 94.4%), and less likely to have smoked in pregnancy (e.g., daily smoking: 4.5% vs 6.3%). Fathers in the analytic sample were also more likely to have stopped smoking during the pregnancy (20.9% vs 15.8%). There was little of a difference in BMI for mothers, fathers, or children. Children in the analytic sample had slightly lower depressive symptoms (mean SMFQ = 1.81 vs 1.91),anxiety symptoms (mean SCARED: 1.04 vs 1.00),and ADHD symptoms (mean RS-DBD ADHD: 8.4 vs 8.7). Descriptive characteristics of the full MoBa sample are in *Appendix 1—table 1*.

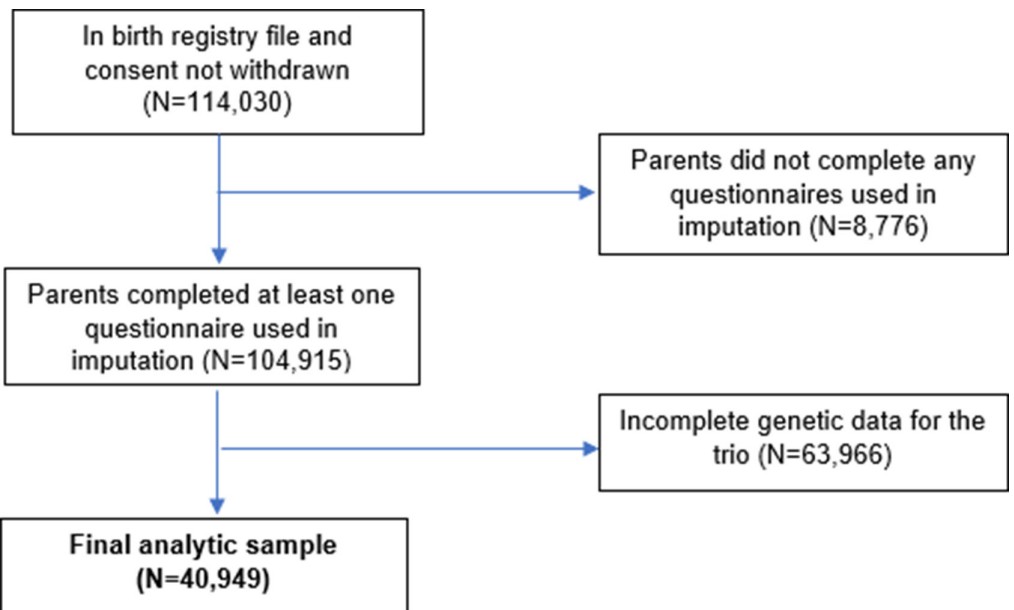

**Appendix 1—figure 1.** Flow chart of inclusion and exclusion of MoBa participants into the study sample.

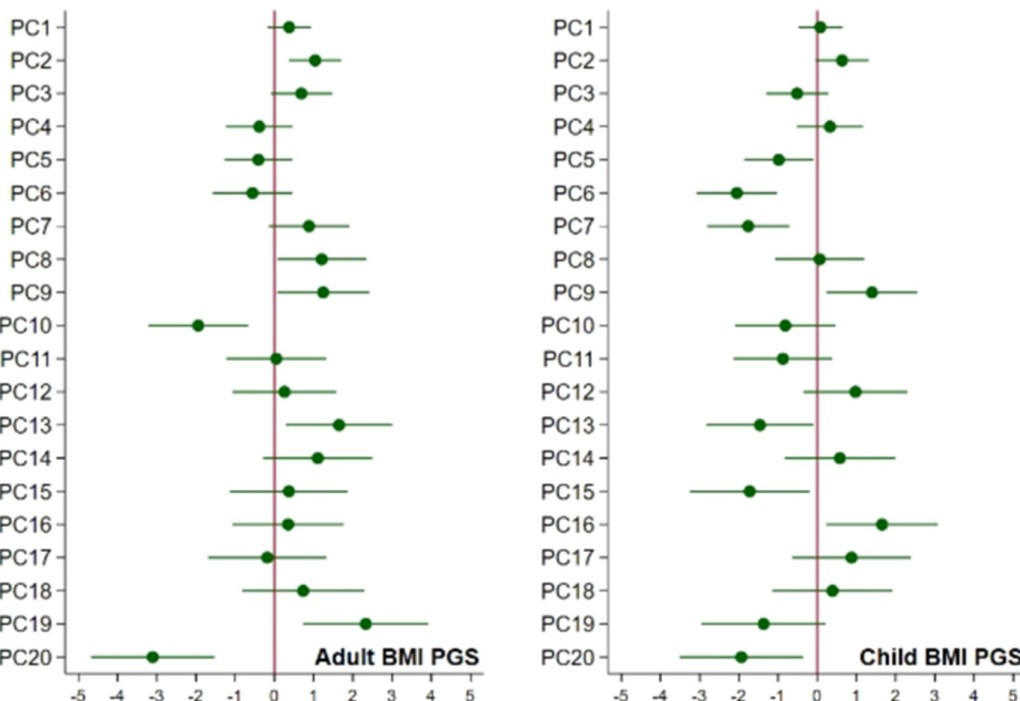

**Appendix 1—figure 2.** Associations of child's BMI polygenic scores with ancestry. Associations of the child's polygenic scores for BMI and the child's principal components of ancestry, adjusted for the child's genotyping centre and chip.

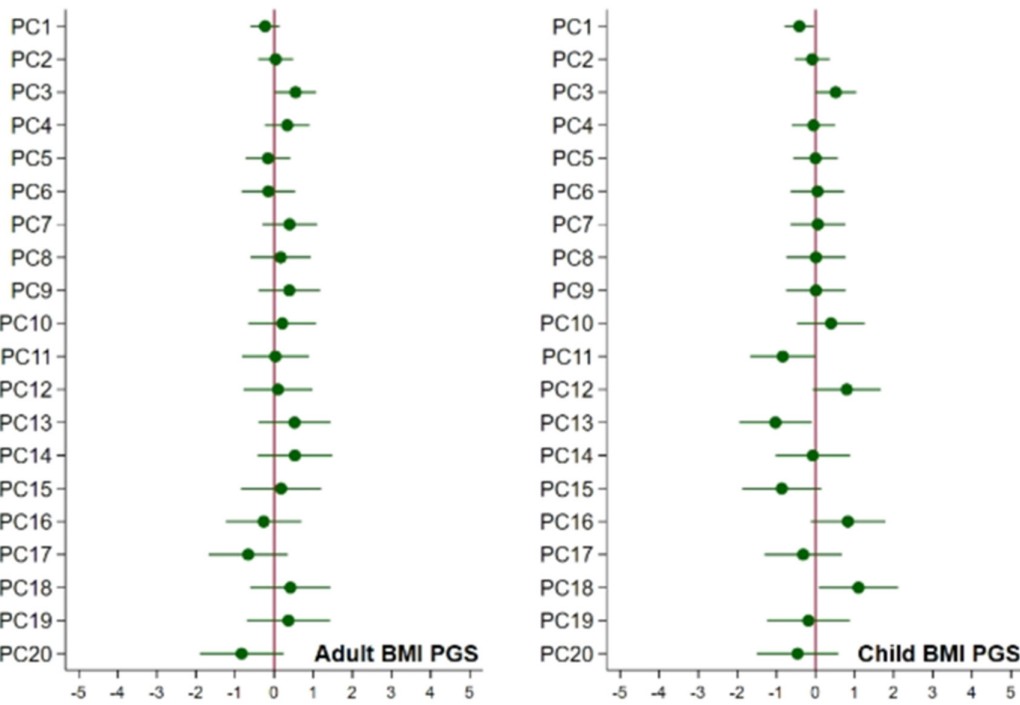

**Appendix 1—figure 3.** Associations of child's BMI polygenic scores with ancestry, adjusted for parental polygenic scores. Associations of the child's polygenic scores for BMI and the child's principal components of ancestry, adjusted for the child's genotyping centre and chip and the parents' polygenic scores.

**Appendix 1—table 1.** Descriptive statistics of full MoBa sample (N=113,691)*.

| Continuous variables | mean | SD | N obs |
|---|---|---|---|
| Maternal age at birth | 30.1 | 4.6 | 113,691 |
| Paternal age at birth | 32.7 | 5.5 | 113,172 |
| Maternal depressive/anxiety symptoms, based on 5 items from the Hopkins Symptoms Checklist-25 (SCL-25)[†] | 1.3 | 2.0 | 100,570 |
| Paternal depressive/anxiety symptoms, based on 8 items from the Hopkins Symptoms Checklist-25 (SCL-25)[‡] | 1.2 | 2.1 | 77,018 |
| Maternal ADHD symptoms: adult ADHD self-report scale[§] | 6.6 | 3.5 | 56,255 |
| Paternal ADHD symptoms: adult ADHD self-report scale[¶] | 8.2 | 3.2 | 34,425 |
| Maternal pre-pregnancy BMI (kg/m$^2$) | 24.0 | 4.1 | 100,060 |
| Paternal BMI (kg/m$^2$) | 25.9 | 3.3 | 79,536 |
| Child's BMI at age 8 (kg/m$^2$) | 16.2 | 2.0 | 36,894 |
| Child depressive symptoms age 8: Short Mood and Feelings Questionnaire (SMFQ)** | 1.9 | 2.4 | 43,065 |
| Child anxiety symptoms age 8: Screen for Child Anxiety Related Disorders (SCARED)[††] | 1.0 | 1.2 | 43,298 |
| Child ADHD symptoms age 8: Parent/Teacher Rating Scale for Disruptive Behaviour Disorders (RS-DBD)[‡‡] | 8.5 | 7.2 | 43,238 |

*Appendix 1—table 1 Continued on next page*

*Appendix 1—table 1 Continued*

| Continuous variables | mean | SD | N obs |
|---|---|---|---|
| Child ADHD symptoms (inattention) age 8: Parent/Teacher Rating Scale for Disruptive Behaviour Disorders (RS-DBD)[§§] | 5.0 | 4.1 | 43,194 |
| Child ADHD symptoms (hyperactivity) age 8: Parent/Teacher Rating Scale for Disruptive Behaviour Disorders (RS-DBD)[§§] | 3.6 | 3.9 | 43,177 |

| Categorical variables | Category | % | N obs |
|---|---|---|---|
| Child's sex | male | 51.3 | 113,477 |
| | female | 48.7 | |
| maternal educational qualifications | 9 year elementary education | 2.8 | 101,020 |
| | Up to 2 years further education | 5.2 | |
| | Further education: vocational | 12.8 | |
| | Further education: general studies, sixth form | 15.7 | |
| | Higher education: college/ university, up to 4 years | 40.9 | |
| | Higher education: college/ university, over 4 years | 22.6 | |
| paternal educational qualifications | 9 year elementary education | 4.8 | 100,891 |
| | Up to 2 years further education | 6.6 | |
| | Further education: vocational | 26.2 | |
| | Further education: general studies, sixth form | 13.2 | |
| | Higher education: college/ university, up to 4 years | 26.5 | |
| | Higher education: college/ university, over 4 years | 22.7 | |
| maternal parity at child's birth | 0 | 44.0 | 113,691 |
| | 1 | 35.9 | |
| | 2 | 15.6 | |
| | 3 | 3.4 | |
| | 4+ | 1.1 | |
| Mother's marital status at birth | Married/registered partner | 95.5 | 113,691 |
| | single | 4.5 | |
| Mother's smoked during pregnancy | never | 49.9 | 101,373 |
| | Stopped before week 17 | 41.7 | |
| | Currently, sometimes | 2.8 | |
| | Currently, daily | 5.6 | |

*All participants in the birth registry file who had not withdrawn consent.
[†]Possible range: 0–15.
[‡]Possible range: 0–24.
[§]Possible range: 0–24.
[¶]Possible range: 0–24.
**Possible range: 0-26.
[††]Possible range: 0-10.
[‡‡]Possible range: 0–54.
[§§]Possible range: 0–27.

**Appendix 1—table 2.** Descriptive statistics of analytic sample (N=40,949)*, unimputed data.

| Continuous variables | mean | SD | N obs |
|---|---|---|---|
| Maternal age at birth | 30.2 | 4.4 | 40,949 |
| Paternal age at birth | 32.6 | 5.1 | 40,945 |
| Maternal depressive/anxiety symptoms, based on 5 items from the Hopkins Symptoms Checklist-25 (SCL-25)† | 1.2 | 1.9 | 39,647 |
| Paternal depressive/anxiety symptoms, based on 8 items from the Hopkins Symptoms Checklist-25 (SCL-25)‡ | 1.1 | 2.1 | 38,050 |
| Maternal ADHD symptoms: adult ADHD self-report scale§ | 6.5 | 3.4 | 24,192 |
| Paternal ADHD symptoms: adult ADHD self-report scale¶ | 8.2 | 3.1 | 16,348 |
| Maternal pre-pregnancy BMI (kg/m²) | 24.0 | 4.1 | 39,323 |
| Paternal BMI (kg/m²) | 25.9 | 3.2 | 39,405 |
| Child's BMI at age 8 (kg/m²) | 16.2 | 2.0 | 16,144 |
| Child depressive symptoms age 8: Short Mood and Feelings Questionnaire (SMFQ)** | 1.8 | 2.4 | 18,747 |
| Child anxiety symptoms age 8: Screen for Child Anxiety Related Disorders (SCARED)†† | 1.0 | 1.2 | 18,834 |
| Child ADHD symptoms age 8: Parent/Teacher Rating Scale for Disruptive Behaviour Disorders (RS-DBD)‡‡ | 8.4 | 7.1 | 18,813 |
| Child ADHD symptoms (inattention) age 8: Parent/Teacher Rating Scale for Disruptive Behaviour Disorders (RS-DBD)§§ | 4.9 | 4.0 | 18,794 |
| Child ADHD symptoms (hyperactivity) age 8: Parent/Teacher Rating Scale for Disruptive Behaviour Disorders (RS-DBD)§§ | 3.5 | 3.8 | 18,787 |

| Categorical variables | Category | % | N obs |
|---|---|---|---|
| Child's sex | male | 51.0 | 40,949 |
| | female | 48.9 | |
| maternal educational qualifications | 9 year elementary education | 2.0 | 39,569 |
| | Up to 2 years further education | 4.1 | |
| | Further education: vocational | 12.1 | |
| | Further education: general studies, sixth form | 14.9 | |
| | Higher education: college/university, up to 4 years | 42.8 | |
| | Higher education: college/university, over 4 years | 24.8 | |
| paternal educational qualifications | 9 year elementary education | 3.4 | 40,062 |
| | Up to 2 years further education | 5.6 | |
| | Further education: vocational | 25.1 | |
| | Further education: general studies, sixth form | 12.9 | |
| | Higher education: college/university, up to 4 years | 28.5 | |
| | Higher education: college/university, over 4 years | 24.4 | |

*Appendix 1—table 2 Continued on next page*

*Appendix 1—table 2 Continued*

| Continuous variables | | mean | SD | N obs |
|---|---|---|---|---|
| maternal parity at child's birth | 0 | | 46.8 | 40,949 |
| | 1 | | 35.7 | |
| | 2 | | 14.0 | |
| | 3 | | 2.7 | |
| | 4+ | | 0.7 | |
| Mother's marital status at birth | Married/registered partner | | 97.4 | 40,949 |
| | single | | 2.6 | |
| Mother's smoked during pregnancy | never | | 51.1 | 40,118 |
| | Stopped before week 17 | | 42.0 | |
| | Currently, sometimes | | 2.4 | |
| | Currently, daily | | 4.5 | |

*The reasons for exclusions and numbers in each case are shown in ***Appendix 1—figure 1***.

†Possible range: 0–15.

‡Possible range: 0–24.

§Possible range: 0–24.

¶Possible range: 0–24.

**Possible range: 0-26.

††Possible range: 0–10.

‡‡Possible range: 0–54.

§§Possible range: 0–27.

**Appendix 1—table 3.** BMI and symptoms of depression, anxiety, and ADHD at age 8 in MoBa, adult BMI PGS (N=40,949)*.

| Outcome | | Non-genetic estimate† | | | MR estimate‡ | | | Within-families MR estimate‡ | | |
|---|---|---|---|---|---|---|---|---|---|---|
| | | Beta (per 5 kg/m²) | CI | p | Beta (per 5 kg/m²) | CI | p | Beta (per 5 kg/m²) | CI | p |
| Depressive symptoms: standardized SMFQ§ score | Child BMI | 0.05 | 0.01,0.09 | 0.02 | 0.45 | 0.26,0.64 | <0.001 | 0.26 | −0.01,0.52 | 0.06 |
| | Mother's BMI | 0.05 | 0.03,0.07 | <0.001 | | | | 0.11 | 0.02,0.19 | 0.01 |
| | Father's BMI | 0.01 | −0.02,0.03 | 0.67 | | | | 0.02 | −0.09,0.13 | 0.71 |
| Anxiety symptoms: standardized SCARED¶ score | Child BMI | −0.07 | −0.11,−0.03 | 0.001 | −0.06 | −0.25,0.12 | 0.51 | 0.01 | −0.25,0.26 | 0.96 |
| | Mother's BMI | 0.01 | −0.01,0.03 | 0.47 | | | | −0.03 | −0.11,0.05 | 0.49 |
| | Father's BMI | −0.00 | −0.02,0.02 | 0.89 | | | | −0.02 | −0.13,0.09 | 0.72 |
| ADHD symptoms: standardized RS-DBD** score, ADHD items | Child BMI | −0.07 | −0.11,−0.03 | 0.001 | 0.35 | 0.17,0.53 | <0.001 | 0.36 | 0.09,0.63 | 0.009 |
| | Mother's BMI | 0.04 | 0.02,0.06 | <0.001 | | | | 0.00 | −0.08,0.09 | 0.97 |
| | Father's BMI | 0.02 | −0.00,0.04 | 0.10 | | | | −0.01 | −0.12,0.09 | 0.80 |
| ADHD-inattention symptoms: standardized RS-DBD** score, inattention items | Child BMI | −0.06 | −0.10,−0.02 | 0.006 | 0.32 | 0.14,0.49 | <0.001 | 0.38 | 0.12,0.65 | 0.005 |
| | Mother's BMI | 0.05 | 0.03,0.07 | <0.001 | | | | 0.01 | −0.08,0.09 | 0.86 |
| | Father's BMI | 0.02 | −0.00,0.05 | 0.06 | | | | −0.06 | −0.17,0.04 | 0.24 |
| ADHD-hyperactivity symptoms: standardized RS-DBD** score, hyperactivity items | Child BMI | −0.06 | −0.10,−0.02 | 0.002 | 0.31 | 0.13,0.49 | 0.001 | 0.27 | −0.00,0.54 | 0.05 |
| | Mother's BMI | 0.03 | 0.01,0.05 | 0.005 | | | | −0.00 | −0.09,0.08 | 0.92 |
| | Father's BMI | 0.01 | −0.01,0.04 | 0.32 | | | | 0.04 | −0.07,0.15 | 0.47 |

*Coefficients represent S.D. change in symptoms per 5 kg/m² increase in BMI.

†Phenotypic models adjust for the child's sex and birth year, the mother's parity at the child's birth, and the mother's and father's: educational qualifications, depressive/anxiety and ADHD symptoms, and smoking status during pregnancy. They also adjust for the child's, mother's, and father's genotyping centre, genotyping chip, and first 20 principal components of ancestry.

‡Genetic models adjust for the child's sex and birth year and the child's, mother's, and father's genotyping centre, genotyping chip, and first 20 principal components of ancestry.

§Short Mood and Feelings Questionnaire.

¶Screen for Child Anxiety Related Disorders.

**Parent/Teacher Rating Scale for Disruptive Behaviour Disorders.

**Appendix 1—table 4.** BMI and symptoms of depression, anxiety, and ADHD at age 8 in MoBa, childhood body size PGS (N=40,949)*.

| Outcome | | Non-genetic estimate† | | | MR estimate ‡ | | | Within-families MR estimate ‡ | | |
|---|---|---|---|---|---|---|---|---|---|---|
| | | Beta (per 5 kg/m²) | CI | p | Beta (per 5 kg/m²) | CI | p | Beta (per 5 kg/m²) | CI | p |
| Depressive symptoms: standardized SMFQ§ score | Child BMI | 0.05 | 0.01,0.09 | 0.02 | 0.08 | -0.07,0.22 | 0.29 | 0.02 | -0.20,0.23 | 0.88 |
| | Mother's BMI | 0.05 | 0.03,0.07 | <0.001 | | | | 0.02 | -0.09,0.14 | 0.67 |
| | Father's BMI | 0.01 | -0.02,0.03 | 0.67 | | | | 0.05 | -0.10,0.19 | 0.51 |
| Anxiety symptoms: standardized SCARED¶ score | Child BMI | -0.07 | -0.11,-0.03 | 0.001 | -0.04 | -0.18,0.11 | 0.62 | 0.02 | -0.18,0.23 | 0.83 |
| | Mother's BMI | 0.01 | -0.01,0.03 | 0.47 | | | | -0.02 | -0.12,0.09 | 0.78 |
| | Father's BMI | -0.00 | -0.02,0.02 | 0.89 | | | | -0.06 | -0.19,0.08 | 0.42 |
| ADHD symptoms: standardized RS-DBD** score, ADHD items | Child BMI | -0.07 | -0.11,-0.03 | 0.001 | -0.07 | -0.21,0.07 | 0.35 | -0.03 | -0.22,0.17 | 0.80 |
| | Mother's BMI | 0.04 | 0.02,0.06 | <0.001 | | | | -0.03 | -0.12,0.07 | 0.62 |
| | Father's BMI | 0.02 | -0.00,0.04 | 0.10 | | | | -0.02 | -0.15,0.11 | 0.77 |
| ADHD-inattention symptoms: standardized RS-DBD** score, inattention items | Child BMI | -0.06 | -0.10,-0.02 | 0.006 | -0.04 | -0.18,0.11 | 0.63 | -0.05 | -0.25,0.14 | 0.59 |
| | Mother's BMI | 0.05 | 0.03,0.07 | <0.001 | | | | 0.03 | -0.07,0.13 | 0.61 |
| | Father's BMI | 0.02 | -0.00,0.05 | 0.06 | | | | -0.01 | -0.14,0.12 | 0.86 |
| ADHD-hyperactivity symptoms: standardized RS-DBD** score, hyperactivity items | Child BMI | -0.06 | -0.10,-0.02 | 0.002 | -0.08 | -0.22,0.05 | 0.24 | 0.01 | -0.20,0.22 | 0.95 |
| | Mother's BMI | 0.03 | 0.01,0.05 | 0.005 | | | | -0.07 | -0.18,0.03 | 0.18 |
| | Father's BMI | 0.01 | -0.01,0.04 | 0.32 | | | | -0.02 | -0.16,0.11 | 0.74 |

*Coefficients represent S.D. change in symptoms per 5 kg/m² increase in BMI.

†Phenotypic models adjust for the child's sex and birth year, the mother's parity at the child's birth, and the mother's and father's: educational qualifications, depressive/anxiety and ADHD symptoms, and smoking status during pregnancy. They also adjust for the child's, mother's, and father's genotyping centre, genotyping chip, and first 20 principal components of ancestry.

‡Genetic models adjust for the child's sex and birth year and the child's, mother's, and father's genotyping centre, genotyping chip, and first 20 principal components of ancestry.

§Short Mood and Feelings Questionnaire.

¶Screen for Child Anxiety Related Disorders.

**Parent/Teacher Rating Scale for Disruptive Behaviour Disorders

**Appendix 1—table 5.** Associations of phenotypes and polygenic scores within parental pairs*.

**Phenotypes: regression of father's BMI, depressive symptoms, and ADHD symptoms on mother's phenotypes**

| | Father's: BMI | Father's: Depressive symptoms | Father's: ADHD symptoms |
|---|---|---|---|
| | Beta (95% CI), p | Beta (95% CI), p | Beta (95% CI), p |
| Mother's: BMI | 0.23 (0.22,0.25) p<0.001 | 0.01 (-0.00,0.02) p=0.09 | 0.03 (0.02,0.05) p<0.001 |
| Mother's: Depressive symptoms | 0.00 (-0.01,0.01), p=0.85 | 0.18 (0.16,0.20) p<0.001 | 0.10 (0.09,0.12), P<0.001 |
| Mother's: ADHD symptoms | 0.01 (-0.02,0.01) p=0.32 | 0.05 (0.05,0.06) p<0.001 | 0.11 (0.09,0.13) p<0.001 |

*Polygenic scores: regression of father's PGS for BMI, depression, and ADHD: regression of father's PGS on mother's PGS*

| | Father's: Adult BMI PGS | Father's: Childhood body size PGS | Father's: Depression PGS | Father's: ADHD PGS |
|---|---|---|---|---|
| | Beta (95% CI), p | Beta (95% CI), p | Beta (95% CI), p | Beta (95% CI) |
| Mother's: Adult BMI PGS | 0.01 (0.00,0.02), p=0.02 | 0.01 (0.00,0.02), p=0.008 | -0.00 (-0.01,0.01), p=0.62 | -0.00 (-0.01,0.01), p=0.49 |
| Mother's: Childhood body size PGS | 0.01 (-0.00,0.02), p=0.10 | 0.01 (-0.00,0.02), p=0.15 | 0.00 (-0.00,0.01), p=0.33 | 0.01 (0.00,0.02), p=0.03 |
| Mother's: Depression PGS | -0.01 (-0.02–0.00), p=0.11 | -0.00 (-0.01,0.01), p=0.76 | -0.00 (-0.01,0.01), p=0.44 | -0.00 (-0.01,0.01), p=0.47 |
| Mother's: ADHD PGS | -0.00 (-0.01,0.01), p=0.58 | 0.01 (-0.00,0.02), p=0.24 | -0.00 (-0.01,0.01), p=0.42 | -0.00 (-0.01,0.01), p=0.93 |

*All models adjusted for the first 20 principal components of ancestry, genotyping centre and genotyping chip of the mother and father.

**Appendix 1—table 6.** BMI and symptoms of depression, anxiety, and ADHD at age 8 in MoBa, adult BMI PGS, log-transformed outcomes (N=40,949)*.

| Outcome | | Non-genetic estimate† | | | MR estimate‡ | | | Within-families MR estimate‡ | | |
|---|---|---|---|---|---|---|---|---|---|---|
| | | Beta (per 5 kg/m²) | CI | p | Beta (per 5 kg/m²) | CI | p | Beta (per 5 kg/m²) | CI | p |
| Depressive symptoms: log-transformed SMFQ§ score | Child BMI | 0.03 | 0.01,0.06 | 0.02 | 0.31 | 0.18,0.45 | <0.001 | 0.16 | -0.04,0.35 | 0.12 |
| | Mother's BMI | 0.04 | 0.03,0.05 | <0.001 | | | | 0.08 | 0.02,0.14 | 0.01 |
| | Father's BMI | 0.01 | -0.01,0.02 | 0.56 | | | | 0.03 | -0.05,0.10 | 0.52 |
| Anxiety symptoms: log-transformed SCARED¶ score | Child BMI | -0.03 | -0.05,-0.01 | 0.001 | -0.04 | -0.14,0.05 | 0.38 | -0.00 | -0.14,0.13 | 0.97 |
| | Mother's BMI | 0.00 | -0.01,0.01 | 0.60 | | | | -0.02 | -0.06,0.02 | 0.35 |
| | Father's BMI | -0.00 | -0.01,0.01 | 1.00 | | | | -0.01 | -0.06,0.05 | 0.79 |
| ADHD symptoms: log-transformed RS-DBD** score | Child BMI | -0.05 | -0.08,-0.02 | 0.002 | 0.27 | 0.13,0.40 | <0.001 | 0.28 | 0.08,0.49 | 0.006 |
| | Mother's BMI | 0.03 | 0.02,0.05 | <0.001 | | | | 0.00 | -0.06,0.07 | 0.90 |
| | Father's BMI | 0.02 | -0.00,0.03 | 0.07 | | | | -0.02 | -0.10,0.06 | 0.66 |
| ADHD-inattention symptoms: log-transformed RS-DBD** score, inattention items | Child BMI | -0.04 | -0.07,-0.01 | 0.007 | 0.22 | 0.09,0.34 | 0.001 | 0.29 | 0.10,0.47 | 0.003 |
| | Mother's BMI | 0.03 | 0.02,0.05 | <0.001 | | | | 0.00 | -0.06,0.07 | 0.92 |
| | Father's BMI | 0.02 | 0.00,0.03 | 0.05 | | | | -0.06 | -0.14,0.02 | 0.13 |
| ADHD-hyperactivity symptoms: log-transformed RS-DBD** score, hyperactivity items | Child BMI | -0.05 | -0.08,-0.01 | 0.004 | 0.24 | 0.10,0.39 | 0.001 | 0.18 | -0.04,0.40 | 0.10 |
| | Mother's BMI | 0.02 | 0.00,0.04 | 0.01 | | | | 0.00 | -0.07,0.07 | 1.00 |
| | Father's BMI | 0.01 | -0.01,0.03 | 0.25 | | | | 0.05 | -0.04,0.14 | 0.29 |

*Coefficients represent change in symptoms, log-transformed after adding 1, per 5 kg/m² increase in BMI.

†Phenotypic models adjust for the child's sex and birth year, the mother's parity at the child's birth, and the mother's and father's: educational qualifications, depressive/anxiety and ADHD symptoms, and smoking status during pregnancy. They also adjust for the child's, mother's, and father's genotyping centre, genotyping chip, and first 20 principal components of ancestry.

‡Genetic models adjust for the child's sex and birth year and the child's, mother's, and father's genotyping centre, genotyping chip, and first 20 principal components of ancestry.

§Short Mood and Feelings Questionnaire.

¶Screen for Child Anxiety Related Disorders.

**Parent/Teacher Rating Scale for Disruptive Behaviour Disorders.

**Appendix 1—table 7.** BMI and symptoms of depression, anxiety, and ADHD at age 8 in MoBa, childhood body size PGS, log-transformed outcomes (N=40,949)*.

| Outcome | | Non-genetic estimate[†] | | | MR estimate[‡] | | | Within-families MR estimate[‡] | | |
|---|---|---|---|---|---|---|---|---|---|---|
| | | Beta (per 5 kg/m²) | CI | p | Beta (per 5 kg/m²) | CI | p | Beta (per 5 kg/m²) | CI | p |
| Depressive symptoms: log-transformed SMFQ[§] score | Child BMI | 0.03 | 0.01,0.06 | 0.02 | 0.07 | −0.04,0.17 | 0.20 | 0.02 | −0.14,0.17 | 0.84 |
| | Mother's BMI | 0.04 | 0.03,0.05 | <0.001 | | | | 0.02 | −0.06,0.10 | 0.67 |
| | Father's BMI | 0.01 | −0.01,0.02 | 0.56 | | | | 0.05 | −0.06,0.15 | 0.39 |
| Anxiety symptoms: log-transformed SCARED[¶] score | Child BMI | −0.03 | −0.05,−0.01 | 0.001 | −0.03 | −0.10,0.05 | 0.49 | 0.01 | −0.10,0.12 | 0.89 |
| | Mother's BMI | 0.00 | −0.01,0.01 | 0.60 | | | | −0.01 | −0.07,0.04 | 0.69 |
| | Father's BMI | −0.00 | −0.01,0.01 | 1.00 | | | | −0.03 | −0.10,0.04 | 0.43 |
| ADHD symptoms: log-transformed RS-DBD** score | Child BMI | −0.05 | −0.08,−0.02 | 0.002 | −0.06 | −0.17,0.05 | 0.30 | −0.03 | −0.19,0.13 | 0.70 |
| | Mother's BMI | 0.03 | 0.02,0.05 | <0.001 | | | | −0.02 | −0.10,0.06 | 0.68 |
| | Father's BMI | 0.02 | −0.00,0.03 | 0.07 | | | | −0.01 | −0.11,0.09 | 0.79 |
| ADHD-inattention symptoms: log-transformed RS-DBD** score, inattention items | Child BMI | −0.04 | −0.07,−0.01 | 0.007 | −0.03 | −0.14,0.07 | 0.52 | −0.05 | −0.19,0.10 | 0.52 |
| | Mother's BMI | 0.03 | 0.02,0.05 | <0.001 | | | | 0.02 | −0.06,0.09 | 0.64 |
| | Father's BMI | 0.02 | 0.00,0.03 | 0.05 | | | | −0.01 | −0.10,0.09 | 0.87 |
| ADHD-hyperactivity symptoms: log-transformed RS-DBD** score, hyperactivity items | Child BMI | −0.05 | −0.08,−0.01 | 0.004 | −0.07 | −0.19,0.04 | 0.21 | −0.00 | −0.18,0.17 | 0.98 |
| | Mother's BMI | 0.02 | 0.00,0.04 | 0.01 | | | | −0.06 | −0.15,0.03 | 0.22 |
| | Father's BMI | 0.01 | −0.01,0.03 | 0.25 | | | | −0.02 | −0.13,0.09 | 0.73 |

*Coefficients represent change in symptoms, log-transformed after adding 1, per 5 kg/m² increase in BMI.

[†]Phenotypic models adjust for the child's sex and birth year, the mother's parity at the child's birth, and the mother's and father's: educational qualifications, depressive/anxiety and ADHD symptoms, and smoking status during pregnancy. They also adjust for the child's, mother's, and father's genotyping centre, genotyping chip, and first 20 principal components of ancestry.

[‡]Genetic models adjust for the child's sex and birth year and the child's, mother's, and father's genotyping centre, genotyping chip, and first 20 principal components of ancestry.

[§]Short Mood and Feelings Questionnaire.

[¶]Screen for Child Anxiety Related Disorders.

**Parent/Teacher Rating Scale for Disruptive Behaviour Disorders.

**Appendix 1—table 8.** Robustness checks based on SNP-specific associations with child's BMI* and outcomes: SNPs in adult BMI polygenic score.

| For classic MR models | Inverse-variance weighted | | MR-Egger: slope | | MR-Egger: intercept | | MR-Median | | MR-Modal | |
|---|---|---|---|---|---|---|---|---|---|---|
| | Beta | p | Beta | p | Beta | p | Beta | p | Beta | p |
| Depressive symptoms: standardized SMFQ† score | 0.12 | <0.001 | 0.09 | 0.18 | 0.00 | 0.56 | 0.11 | <0.001 | 0.07 | 0.42 |
| Anxiety symptoms: standardized SCARED‡ score | −0.02 | 0.13 | −0.05 | 0.45 | 0.00 | 0.61 | −0.01 | 0.87 | 0.01 | 0.92 |
| ADHD symptoms: standardized RS-DBD§ score | 0.10 | <0.001 | 0.02 | 0.77 | 0.00 | 0.19 | 0.09 | 0.01 | 0.07 | 0.40 |
| ADHD symptoms (inattention): standardized RS-DBD§ score | 0.09 | <0.001 | −0.02 | 0.73 | 0.00 | 0.07 | 0.09 | 0.01 | −0.02 | 0.81 |
| ADHD symptoms (hyperactivity): standardized RS-DBD§ score | 0.08 | <0.001 | 0.05 | 0.40 | 0.00 | 0.61 | 0.08 | 0.02 | 0.05 | 0.55 |

| For within-families MR models | Inverse-variance weighted | | MR-Egger: slope | | MR-Egger: intercept | | MR-Median | | MR-Modal | |
|---|---|---|---|---|---|---|---|---|---|---|
| | Beta | p | Beta | p | Beta | p | Beta | p | Beta | p |
| Depressive symptoms: standardized SMFQ† score | 0.06 | <0.001 | 0.04 | 0.67 | 0.00 | 0.77 | 0.07 | 0.16 | 0.02 | 0.83 |
| Anxiety symptoms: standardized SCARED‡ score | 0.00 | 1.00 | 0.05 | 0.58 | 0.00 | 0.57 | 0.03 | 0.57 | 0.03 | 0.76 |
| ADHD symptoms: standardized RS-DBD§ score | 0.09 | <0.001 | 0.07 | 0.41 | 0.00 | 0.80 | 0.09 | 0.07 | 0.13 | 0.24 |
| ADHD symptoms (inattention): standardized RS-DBD§ score | 0.10 | <0.001 | 0.00 | 0.97 | 0.00 | 0.21 | 0.10 | 0.03 | 0.05 | 0.63 |
| ADHD symptoms (hyperactivity): standardized RS-DBD§ score | 0.07 | <0.001 | 0.13 | 0.13 | 0.00 | 0.44 | 0.07 | 0.14 | 0.06 | 0.56 |

*In the main analyses, all coefficients are expressed in terms of S.D. change in symptoms per 5 kg/m² increase in BMI. In robustness checks, SNP-exposure associations were taken directly from the relevant GWAS. Coefficients above for BMI-outcome associations are therefore on the scale of kg/m².

†Short Mood and Feelings Questionnaire.

‡Screen for Child Anxiety Related Disorders.

§Parent/Teacher Rating Scale for Disruptive Behaviour Disorders.

**Appendix 1—table 9.** Robustness checks based on SNP-specific associations with child's BMI* and outcomes: SNPs in childhood body size polygenic score.

| For classic MR models | Inverse-variance weighted | | MR-Egger: slope | | MR-Egger: intercept | | MR-Median | | MR-Modal | |
|---|---|---|---|---|---|---|---|---|---|---|
| | Beta | p | Beta | p | Beta | p | Beta | p | Beta | p |
| Depressive symptoms: standardized SMFQ† score | 0.06 | 0.03 | 0.10 | 0.30 | 0.00 | 0.57 | 0.05 | 0.52 | 0.04 | 0.67 |
| Anxiety symptoms: standardized SCARED‡ score | −0.02 | 0.40 | 0.01 | 0.90 | 0.00 | 0.66 | −0.02 | 0.74 | −0.01 | 0.95 |
| ADHD symptoms: standardized RS-DBD§ score | −0.04 | 0.14 | −0.02 | 0.85 | 0.00 | 0.75 | −0.06 | 0.36 | −0.05 | 0.65 |
| ADHD symptoms (inattention): standardized RS-DBD§ score | −0.02 | 0.47 | −0.04 | 0.64 | 0.00 | 0.81 | −0.08 | 0.28 | −0.08 | 0.42 |
| ADHD symptoms (hyperactivity): standardized RS-DBD§ score | −0.05 | 0.05 | 0.01 | 0.90 | 0.00 | 0.42 | −0.05 | 0.48 | −0.04 | 0.68 |

| For within-families MR models | Inverse-variance weighted | | MR-Egger: slope | | MR-Egger: intercept | | MR-Median | | MR-Modal | |
|---|---|---|---|---|---|---|---|---|---|---|
| | Beta | p | Beta | p | Beta | p | Beta | p | Beta | p |
| Depressive symptoms: standardized SMFQ† score | 0.02 | 0.59 | 0.05 | 0.70 | 0.00 | 0.75 | 0.03 | 0.76 | 0.04 | 0.79 |
| Anxiety symptoms: standardized SCARED‡ score | 0.02 | 0.60 | 0.07 | 0.64 | 0.00 | 0.69 | 0.04 | 0.66 | 0.05 | 0.74 |
| ADHD symptoms: standardized RS-DBD§§ score | −0.02 | 0.53 | 0.09 | 0.49 | 0.00 | 0.35 | −0.01 | 0.94 | 0.03 | 0.85 |
| ADHD symptoms (inattention): standardized RS-DBD§ score | −0.04 | 0.28 | 0.06 | 0.65 | 0.00 | 0.41 | −0.04 | 0.68 | 0.01 | 0.95 |
| ADHD symptoms (hyperactivity): standardized RS-DBD§ score | 0.00 | 0.98 | 0.11 | 0.43 | 0.00 | 0.39 | 0.02 | 0.80 | 0.06 | 0.67 |

*In the main analyses, all coefficients are expressed in terms of S.D. change in symptoms per 5 kg/m² increase in BMI. In robustness checks, SNP-exposure associations were taken directly from the relevant GWAS. Coefficients above for BMI-outcome associations are therefore on the scale of kg/m².

†Short Mood and Feelings Questionnaire.

‡Screen for Child Anxiety Related Disorders.

§Parent/Teacher Rating Scale for Disruptive Behaviour Disorders.

**Appendix 1—table 10.** BMI and symptoms of depression, anxiety, and ADHD at age 8 in MoBa, adult BMI PGS, complete-case analysis*.

| Outcome | | Non-genetic estimate† | | | MR estimate ‡ | | | Within-families MR estimate ‡ | | |
|---|---|---|---|---|---|---|---|---|---|---|
| | | Beta (per 5 kg/m²) | CI | p | Beta (per 5 kg/m²) | CI | p | Beta (per 5 kg/m²) | CI | p |
| Depressive symptoms: standardized SMFQ§ score. N=5,158 | Child BMI | 0.08 | 0.00,0.15 | 0.05 | 0.38 | 0.00,0.77 | 0.05 | 0.16 | −0.35,0.66 | 0.55 |
| | Mother's BMI | 0.07 | 0.03,0.11 | <0.001 | | | | 0.10 | −0.06,0.26 | 0.21 |
| | Father's BMI | 0.01 | −0.04,0.05 | 0.79 | | | | 0.04 | −0.15,0.23 | 0.66 |
| Anxiety symptoms: standardized SCARED¶ score. N=5,177 | Child BMI | −0.04 | −0.12,0.03 | 0.26 | −0.15 | −0.54,0.24 | 0.45 | −0.07 | −0.62,0.48 | 0.80 |
| | Mother's BMI | 0.02 | −0.02,0.06 | 0.26 | | | | 0.06 | −0.11,0.22 | 0.48 |
| | Father's BMI | 0.01 | −0.04,0.05 | 0.69 | | | | −0.13 | −0.34,0.08 | 0.24 |
| ADHD symptoms: standardized RS-DBD** score, ADHD items. N=5,174 | Child BMI | −0.03 | −0.11,0.04 | 0.38 | 0.41 | −0.00,0.83 | 0.05 | 0.54 | 0.01,1.08 | 0.04 |
| | Mother's BMI | 0.09 | 0.05,0.12 | <0.001 | | | | −0.01 | −0.16,0.15 | 0.93 |
| | Father's BMI | −0.01 | −0.05,0.03 | 0.60 | | | | −0.09 | −0.29,0.12 | 0.41 |
| ADHD-inattention symptoms: standardized RS-DBD** score, inattention items. N=5,171 | Child BMI | −0.02 | −0.10,0.05 | 0.55 | 0.44 | 0.04,0.85 | 0.03 | 0.73 | 0.20,1.25 | 0.01 |
| | Mother's BMI | 0.09 | 0.05,0.12 | <0.001 | | | | −0.02 | −0.18,0.14 | 0.84 |
| | Father's BMI | −0.00 | −0.05,0.04 | 0.83 | | | | −0.18 | −0.39,0.03 | 0.09 |
| ADHD-hyperactivity symptoms: standardized RS-DBD** score, hyperactivity items. N=5,167 | Child BMI | −0.04 | −0.11,0.03 | 0.29 | 0.28 | −0.13,0.70 | 0.18 | 0.21 | −0.32,0.75 | 0.43 |
| | Mother's BMI | 0.07 | 0.03,0.11 | 0.001 | | | | 0.01 | −0.15,0.17 | 0.87 |
| | Father's BMI | −0.01 | −0.06,0.03 | 0.51 | | | | 0.03 | −0.17,0.24 | 0.75 |

*Coefficients represent S.D. change in symptoms per 5 kg/m² increase in BMI.

†Phenotypic models adjust for the child's sex and birth year, the mother's parity at the child's birth, and the mother's and father's: educational qualifications, depressive/anxiety and ADHD symptoms, and smoking status during pregnancy. They also adjust for the child's, mother's, and father's genotyping centre, genotyping chip, and first 20 principal components of ancestry.

‡Genetic models adjust for the child's sex and birth year and the child's, mother's, and father's genotyping centre, genotyping chip, and first 20 principal components of ancestry.

§Short Mood and Feelings Questionnaire

¶Screen for Child Anxiety Related Disorders

**Parent/Teacher Rating Scale for Disruptive Behaviour Disorders.

**Appendix 1—table 11.** BMI and symptoms of depression, anxiety, and ADHD at age 8 in MoBa, childhood body size PGS, complete-case analysis*.

| Outcome | | Non-genetic estimate† | | | MR estimate‡ | | | Within-families MR estimate‡ | | |
|---|---|---|---|---|---|---|---|---|---|---|
| | | Beta (per 5 kg/m²) | CI | p | Beta (per 5 kg/m²) | CI | p | Beta (per 5 kg/m²) | CI | p |
| Depressive symptoms: standardized SMFQ§ score N=5,158 | Child BMIᵃ | 0.08 | 0.00,0.15 | 0.05 | 0.13 | -0.15,0.42 | 0.37 | 0.10 | -0.34,0.54 | 0.65 |
| | Mother's BMI | 0.07 | 0.03,0.11 | <0.001 | | | | -0.06 | -0.26,0.14 | 0.57 |
| | Father's BMI | 0.01 | -0.04,0.05 | 0.79 | | | | 0.10 | -0.20,0.40 | 0.52 |
| Anxiety symptoms: standardized SCARED¶ score N=5,177 | Child BMI | -0.04 | -0.12,0.03 | 0.26 | -0.03 | -0.32,0.27 | 0.85 | 0.06 | -0.37,0.48 | 0.79 |
| | Mother's BMI | 0.02 | -0.02,0.06 | 0.26 | | | | 0.02 | -0.19,0.24 | 0.85 |
| | Father's BMI | 0.01 | -0.04,0.05 | 0.69 | | | | -0.13 | -0.43,0.18 | 0.42 |
| ADHD symptoms: standardized RS-DBD** score, ADHD items N=5,174 | Child BMI | -0.03 | -0.11,0.04 | 0.38 | 0.00 | -0.29,0.30 | 0.98 | 0.13 | -0.31,0.56 | 0.57 |
| | Mother's BMI | 0.09 | 0.05,0.12 | <0.001 | | | | -0.05 | -0.25,0.15 | 0.61 |
| | Father's BMI | -0.01 | -0.05,0.03 | 0.60 | | | | -0.09 | -0.40,0.22 | 0.56 |
| ADHD-inattention symptoms: standardized RS-DBD** score, inattention items N=5,171 | Child BMI | -0.02 | -0.10,0.05 | 0.55 | -0.00 | -0.29,0.29 | 0.99 | 0.13 | -0.29,0.55 | 0.55 |
| | Mother's BMI | 0.09 | 0.05,0.12 | <0.001 | | | | 0.04 | -0.17,0.24 | 0.72 |
| | Father's BMI | -0.00 | -0.05,0.04 | 0.83 | | | | -0.20 | -0.51,0.11 | 0.20 |
| ADHD-hyperactivity symptoms: standardized RS-DBD** score, hyperactivity items N=5,167 | Child BMI | -0.04 | -0.11,0.03 | 0.29 | 0.00 | -0.30,0.30 | 0.99 | 0.10 | -0.35,0.55 | 0.67 |
| | Mother's BMI | 0.07 | 0.03,0.11 | 0.001 | | | | -0.14 | -0.34,0.06 | 0.18 |
| | Father's BMI | -0.01 | -0.06,0.03 | 0.51 | | | | 0.03 | -0.28,0.34 | 0.84 |

*Coefficients represent S.D. change in symptoms per 5 kg/m² increase in BMI.

†Phenotypic models adjust for the child's sex and birth year, the mother's parity at the child's birth, and the mother's and father's: educational qualifications, depressive/anxiety and ADHD symptoms, and smoking status during pregnancy. They also adjust for the child's, mother's, and father's genotyping centre, genotyping chip, and first 20 principal components of ancestry.

‡Genetic models adjust for the child's sex and birth year and the child's, mother's, and father's genotyping centre, genotyping chip, and first 20 principal components of ancestry.

§Short Mood and Feelings Questionnaire.

¶Screen for Child Anxiety Related Disorders.

**Parent/Teacher Rating Scale for Disruptive Behaviour Disorders.

**Appendix 1—table 12.** Multivariable-adjusted associations* of BMI quintiles with symptoms of depression, anxiety, and ADHD at age 8 in MoBa.

| BMI quintile | Depressive symptoms: standardized SMFQ† score | | Anxiety symptoms: standardized SCARED ‡ score | | ADHD symptoms: standardized RS-DBD § score, ADHD items | | ADHD-inattention symptoms: standardized RS-DBD § score, inattention items | | ADHD-hyperactivity symptoms: standardized RS-DBD § score, hyperactivity items | |
|---|---|---|---|---|---|---|---|---|---|---|
| | Beta ¶ (95% CI) | p | Beta ¶ (95% CI) | p | Beta ¶ (95% CI) | p | Beta ¶ (95% CI) | p | Beta ¶ (95% CI) | p |
| 1 | 0.00 (-0.04,0.04) | 0.99 | 0.04 (0.00,0.09) | 0.04 | 0.05 (0.01,0.09) | 0.01 | 0.05 (0.01,0.09) | 0.02 | 0.04 (-0.00,0.08) | 0.05 |
| 2 | -0.01 (-0.05,0.03) | 0.70 | 0.02 (-0.03,0.06) | 0.47 | 0.02 (-0.02,0.06) | 0.29 | 0.02 (-0.02,0.06) | 0.38 | 0.02 (-0.02,0.06) | 0.30 |
| 3 (ref) | 1 | | 1 | | 1 | | 1 | | 1 | |
| 4 | 0.01 (-0.03,0.05) | 0.62 | -0.03 (-0.07,0.01) | 0.16 | -0.01 (-0.04,0.03) | 0.66 | -0.01 (-0.05,0.02) | 0.54 | -0.00 (-0.04,0.03) | 0.87 |
| 5 | 0.04 (0.00,0.08) | 0.03 | -0.03 (-0.07,0.01) | 0.09 | -0.02 (-0.06,0.02) | 0.27 | -0.01 (-0.05,0.03) | 0.51 | -0.03 (-0.06,0.01) | 0.20 |

*Models adjust for the child's sex and birth year, the mother's parity at the child's birth, and the mother's and father's: educational qualifications, depressive/anxiety and ADHD symptoms, and smoking status during pregnancy. They also adjust for the child's, mother's, and father's genotyping centre, genotyping chip, and first 20 principal components of ancestry.

†Short Mood and Feelings Questionnaire.

‡Screen for Child Anxiety Related Disorders.

§Parent/Teacher Rating Scale for Disruptive Behaviour Disorders.

¶Coefficients represent S.D. difference in symptoms between quintiles of child's BMI.

**Appendix 1—table 13.** BMI and symptoms of depression, anxiety, and ADHD at age 8 in MoBa, adult BMI PGS, genetic models adjusted for parental education (N=40,949)*.

| Outcome | | MR estimate[†] | | | Within-families MR estimate[†] | | |
|---|---|---|---|---|---|---|---|
| | | Beta (per 5 kg/m²) | CI | p | Beta (per 5 kg/m²) | CI | p |
| Depressive symptoms: standardized SMFQ[‡] score | Child BMI | 0.38 | 0.19,0.58 | <0.001 | 0.26 | −0.01,0.53 | 0.06 |
| | Mother's BMI | | | | 0.09 | −0.00,0.17 | 0.05 |
| | Father's BMI | | | | −0.00 | −0.11,0.11 | 0.97 |
| Anxiety symptoms: standardized SCARED[§] score | Child BMI | −0.08 | −0.26,0.11 | 0.43 | 0.01 | −0.25,0.27 | 0.95 |
| | Mother's BMI | | | | −0.04 | −0.12,0.05 | 0.40 |
| | Father's BMI | | | | −0.03 | −0.14,0.08 | 0.64 |
| ADHD symptoms: standardized RS-DBD[¶] score, ADHD items | Child BMI | 0.27 | 0.09,0.45 | 0.003 | 0.37 | 0.10,0.64 | 0.007 |
| | Mother's BMI | | | | −0.03 | −0.12,0.06 | 0.52 |
| | Father's BMI | | | | −0.05 | −0.16,0.06 | 0.37 |
| ADHD-inattention symptoms: standardized RS-DBD[¶] score, inattention items | Child BMI | 0.25 | 0.07,0.42 | 0.007 | 0.39 | 0.13,0.66 | 0.004 |
| | Mother's BMI | | | | −0.02 | −0.11,0.07 | 0.64 |
| | Father's BMI | | | | −0.10 | −0.21,0.01 | 0.08 |
| ADHD-hyperactivity symptoms: standardized RS-DBD[¶] score, hyperactivity items | Child BMI | 0.24 | 0.06,0.42 | 0.009 | 0.27 | 0.00,0.55 | 0.05 |
| | Mother's BMI | | | | −0.03 | −0.12,0.06 | 0.49 |
| | Father's BMI | | | | 0.01 | −0.10,0.12 | 0.85 |

*Coefficients represent S.D. change in symptoms per 5 kg/m² increase in BMI.

[†]Models adjust for the child's sex and birth year and the child's, mother's, and father's genotyping centre, genotyping chip, and father's genotyping centre, genotyping chip, and first 20 principal components of ancestry.

[‡]Short Mood and Feelings Questionnaire.

[§]Screen for Child Anxiety Related Disorders.

[¶]Parent/Teacher Rating Scale for Disruptive Behaviour Disorders.

**Appendix 1—table 14.** BMI and symptoms of depression, anxiety, and ADHD at age 8 in MoBa, childhood body size PGS, genetic models adjusted for parental education (N=40,949)*.

| Outcome | | MR estimate[†] | | | Within-families MR estimate[†] | | |
|---|---|---|---|---|---|---|---|
| | | Beta (per 5 kg/m$^2$) | CI | p | Beta (per 5 kg/m$^2$) | CI | p |
| Depressive symptoms: standardized SMFQ[‡] score | Child BMI | 0.07 | –0.07,0.22 | 0.33 | 0.01 | –0.20,0.23 | 0.90 |
| | Mother's BMI | | | | 0.02 | –0.10,0.13 | 0.76 |
| | Father's BMI | | | | 0.05 | –0.09,0.20 | 0.48 |
| Anxiety symptoms: standardized SCARED[§] score | Child BMI | –0.03 | –0.18,0.11 | 0.65 | 0.03 | –0.18,0.23 | 0.81 |
| | Mother's BMI | | | | –0.02 | –0.13,0.09 | 0.76 |
| | Father's BMI | | | | –0.06 | –0.19,0.08 | 0.44 |
| ADHD symptoms: standardized RS-DBD[¶] score, ADHD items | Child BMI | –0.08 | –0.22,0.06 | 0.27 | –0.03 | –0.23,0.16 | 0.75 |
| | Mother's BMI | | | | –0.03 | –0.13,0.07 | 0.50 |
| | Father's BMI | | | | –0.02 | –0.15,0.11 | 0.79 |
| ADHD-inattention symptoms: standardized RS-DBD[¶] score, inattention items | Child BMI | –0.05 | –0.20,0.10 | 0.52 | –0.06 | –0.25,0.14 | 0.57 |
| | Mother's BMI | | | | 0.02 | –0.08,0.12 | 0.74 |
| | Father's BMI | | | | –0.01 | –0.14,0.12 | 0.87 |
| ADHD-hyperactivity symptoms: standardized RS-DBD[¶] score, hyperactivity items | Child BMI | –0.10 | –0.23,0.04 | 0.18 | –0.00 | –0.21,0.21 | 1.00 |
| | Mother's BMI | | | | –0.08 | –0.19,0.03 | 0.15 |
| | Father's BMI | | | | –0.02 | –0.15,0.12 | 0.77 |

*Coefficients represent S.D. change in symptoms per 5 kg/m$^2$ increase in BMI.

[†]Models adjust for the child's sex and birth year and the child's, mother's, and father's genotyping centre, genotyping chip, and first 20 principal components of ancestry.

[‡]Short Mood and Feelings Questionnaire.

[§]Screen for Child Anxiety Related Disorders.

[¶]Parent/Teacher Rating Scale for Disruptive Behaviour Disorders.

