## [Editor Report]

The manuscript uses genetic effects on BMI to test whether BMI affecxts childhood emotional and behavioural problems: symptoms of depression, anxiety, and attention-deficit and hyperactivity disorder (ADHD) at age 8. By using a within-family design in a large sample of children with genotyped parents in Norway, the study finds that previous estimates of the effect of BMI on childhood emotional and behavioural symptoms may have been overestimated due to confounding with the environment. Larger samples will be needed to determine whether there is a causal effect of BMI on childhood emotional or behavioural problems, and what size it is.

---

## [Decision Letter]

**Decision letter after peer review:**

Thank you for submitting your article "Body mass index and childhood symptoms of depression, anxiety, and attention-deficit hyperactivity disorder: a within-family Mendelian randomization study" for consideration by *eLife*. Your article has been reviewed by 3 peer reviewers, and the evaluation has been overseen by a Reviewing Editor and Ma-Li Wong as the Senior Editor. The following individual involved in review of your submission has agreed to reveal their identity: Daniel Jordan (Reviewer #2).

Essential revisions:

The null result reported by the authors is potentially important to the epidemiology of childhood obesity, and would also be an important practical illustration of the power of recently-developed within-family Mendelian Randomization (MR) approaches. The paper is generally well-written and well presented. However, given that these within-family methods are known to have lower power than traditional MR, confidently reporting a null association requires demonstrating that the study design has sufficient power to detect the expected association. The authors have not sufficiently demonstrated this in the current version of the manuscript. This is the major point that needs to be addressed and all three reviewers provide detailed suggestions on this end. There are also further points raised by the reviewers regarding the multivariate regression analysis and doing a formal statistical comparison between MR and within family MR that need to be addressed.

*Reviewer #1 (Recommendations for the authors):*

1) No direct statistical comparison of 'classic MR' and 'within-family MR' estimates are given. It is hard to know whether there is any statistically sound evidence that the within-family MR estimates are different from the 'classic MR' estimates. It would help to compute the difference between these estimates along with their standard errors. A way to do this would be to derive 'classic MR' estimates from the 'within-family MR' model. The 'classic MR' estimate should be approximately equal to the within-family MR estimate plus the average of the coefficients on the paternal and maternal PGS. (This becomes slightly more complicated when taking assortative mating into account, but this does not seem to be much of an issue here.) Expressing things in this way, the 'classic MR' and 'within-family MR' estimates could both be expressed as linear combinations of the parameters of the within-family MR model, and the standard error of the difference easily computed.

2) It would help to see if there is a non-zero within-family effect of the full childhood and adult BMI PGSs on childhood emotional problems. The authors could use genome-wide summary statistics to construct a PGS using LD-pred or similar tool. While this goes outside of the typical practice of MR, it would be considerably more powerful since the PGS would explain a lot more variance than the PGSs constructed from genome-wide significant SNPs alone. A null result from this analysis would support the authors argument that the classic MR estimate is the result of confounding due to gene-environment correlation.

3) Given the impressive sample size of genotyped trios, the fact that the estimates for within-family MR are still rather imprecise deserves some discussion. Does within-family MR have data requirements that currently cannot be met outside of certain privately held datasets?

*Reviewer #2 (Recommendations for the authors):*

In addition to the broader comment about the need to demonstrate that the study would be powered to detect an association if one is present, I have only one other substantive scientific comment:

1. The multivariate regression analyses control for a series of parental traits, while the MR analyses do not. Meanwhile, the MR analyses control for genetic ancestry (in the form of PCs), while the regression analyses do not. Could the difference in results between multivariate regression and MR come from the difference in the covariates used, rather than the use of the genetic instrument? Is there a good reason to believe that controlling for ancestry would make no difference in the regression analysis, or that controlling for parental traits would make no difference in the MR analysis?

In addition, I have a few comments about presentation that would make the paper easier to read and follow:

2. I find it confusing the use of the phrase "multivariate regression" to describe the observational epidemiology approach, because many MR methods are arguably a form of regression themselves, just using the genetic instrument as an independent variable rather than the observed trait.

3. The Results section reads as though some pieces of it may have been reordered or moved down to the Methods section. This is most glaring at the very beginning, where the MoBa cohort and the analytic sample are not properly introduced. It would benefit from a readthrough to make sure that everything mentioned in the Results section is sufficiently explained there.

4. Most of the Results paragraphs read like listing off statistics, and it's very difficult to tell what conclusions we are meant to draw from each paragraph. Simply adding a concluding sentence to each paragraph would go a long way towards fixing this. It also might be helpful to see p-values in the Results, rather than just estimates and confidence intervals.

5. The MR methods used are not described in very much detail, and the MR methods papers cited in the introduction are not sufficient for me to reconstruct the analyses performed. At a minimum, the methods used in the primary analysis in figures 2-3 and tables 2-3 should be identified by name like the methods used in the robustness analysis in tables 5-6. Ideally the methods and the differences between them should also be described in more detail in the text, as not all readers will necessarily be familiar with the range of MR methods available.

---

## [Author Response]

Essential revisions:The null result reported by the authors is potentially important to the epidemiology of childhood obesity, and would also be an important practical illustration of the power of recently-developed within-family Mendelian Randomization (MR) approaches. The paper is generally well-written and well presented. However, given that these within-family methods are known to have lower power than traditional MR, confidently reporting a null association requires demonstrating that the study design has sufficient power to detect the expected association. The authors have not sufficiently demonstrated this in the current version of the manuscript. This is the major point that needs to be addressed and all three reviewers provide detailed suggestions on this end. There are also further points raised by the reviewers regarding the multivariate regression analysis and doing a formal statistical comparison between MR and within family MR that need to be addressed.

We have addressed the reviewers’ concerns about the power of within-family MR models in several ways. Firstly, the analysis has been completely redone to draw on a larger release of genetic data, resulting in an analytic sample 55% bigger, with corresponding increases in precision. Secondly, various additional changes have been made for more efficient use of data (for example, around the construction of polygenic scores, and using more sources of information on parental education). These changes are specified at the end of this document. Thirdly, we have updated the discussion to clearly acknowledge that in many cases precision of estimates is nevertheless too low to draw firm conclusions about the presence or absence of effects. As discussed in response to a comment by reviewer 1, MoBa is currently the largest individual study in which this approach can be applied. However, new data is becoming available all the time which will allow analyses of this kind within and across studies. In this context, we explore the application of within-family MR with the hope that it will be applied in larger data sources (including via meta-analysis) as more data becomes available.

Additionally, we now provide a formal comparison of the key estimates from classic and within-family MR models in the text.

Reviewer #1 (Recommendations for the authors):1) No direct statistical comparison of 'classic MR' and 'within-family MR' estimates are given. It is hard to know whether there is any statistically sound evidence that the within-family MR estimates are different from the 'classic MR' estimates. It would help to compute the difference between these estimates along with their standard errors. A way to do this would be to derive 'classic MR' estimates from the 'within-family MR' model. The 'classic MR' estimate should be approximately equal to the within-family MR estimate plus the average of the coefficients on the paternal and maternal PGS. (This becomes slightly more complicated when taking assortative mating into account, but this does not seem to be much of an issue here.) Expressing things in this way, the 'classic MR' and 'within-family MR' estimates could both be expressed as linear combinations of the parameters of the within-family MR model, and the standard error of the difference easily computed.

For all outcomes, confidence intervals for classic MR and within-family MR estimates overlap substantially. This is made clear graphically in Figures 1 and 2. However, we have now updated the Results section to comment on z-tests of difference which formally compare estimates from the classic MR and within-family MR models. As noted, estimates of the difference obtained from this method are approximate. However, since the smallest p-value for the difference of the estimates was p=0.26 (for depressive symptoms, using the adult BMI PGS) this will not have affected conclusions. We have also updated the first paragraph of the discussion to explicitly state that ‘estimates from classic MR and within-family MR were consistent for all outcomes’.

2) It would help to see if there is a non-zero within-family effect of the full childhood and adult BMI PGSs on childhood emotional problems. The authors could use genome-wide summary statistics to construct a PGS using LD-pred or similar tool. While this goes outside of the typical practice of MR, it would be considerably more powerful since the PGS would explain a lot more variance than the PGSs constructed from genome-wide significant SNPs alone. A null result from this analysis would support the authors argument that the classic MR estimate is the result of confounding due to gene-environment correlation.

We derived full childhood and adult BMI PGS PRSice-2 (11). SNPs available in both the full GWAS results and MoBa were clumped in MR-Base as above, but no p-value threshold applied. Polygenic scores based on the remaining SNPs (N=14,382 for adult BMI, N=19,877 for childhood body size) were input into PRSice. As expected, R^2^ attributable to these full PGSs (net of PCs and other genetic covariates) were higher than equivalent R^2^ for the versions using the standard p-value threshold, but the difference was modest: the full adult BMI PGS explained 4.3% of the variation in children’s BMI, 8.2% for variation in mothers’ BMI, and 8.1% of the variation in father’s BMI. For the full childhood body size PGS, this was 5.7%, 3.3% and 3.0%.

Results for classic and within-family MR models using these full PGSs are shown in Author response table 1, . For the adult BMI PGS, within-family MR estimates using both versions of the score were consistent for depression (e.g., 0.31, CI:0.06,0.57, p=0.02 with the full PGS, vs. 0.26, CI: -0.01,0.52, p=0.06 for the restricted PGS). For ADHD symptoms, there was a larger point estimate using the full compared to the restricted adult BMI PGS, but confidence intervals overlapped (0.69, CI:0.43,0.95, p<0.001, vs 0.36, CI:0.09,0.63, p=0.009). For anxiety, both classic and within-families MR estimates were larger (and negative) using the full compared to restricted versions of both BMI PGSs. For instance, the within-family MR estimates were -0.34 (CI:-0.59,-0.10) p=0.005 using the full adult BMI PGS, vs 0.01 (CI: -0.25, 0.26) p=0.96 using the restricted adult BMI PGS. Similarly, estimates were -0.24 (CI:-0.43,-0.05), p=0.01 using the full childhood body size PGS, and 0.02 (CI: -0.18,0.23), p=0.83 using the restricted BMI PGS.

**Author response table 1. sa2table1:** BMI and symptoms of depression, anxiety, and ADHD at age 8 in MoBa, using adult and childhood body size PGS with a p-value threshold of 1.

		Adult BMI PGS, p-value threshold of 1^a^	Childhood body size PGS, p-value threshold of 1^a^
		MR estimate^b^	Within-families MR estimate^b^	MR estimate^b^	Within-families MR estimate^b^
		Beta	CI	p	Beta	CI	p	Beta	CI	p	Beta	CI	p
Depressive symptoms: SMFQ^c^ score	Child BMI^a^	0.43	0.26,0.60	<0.001	0.31	0.06,0.57	0.02	0.1	–0.05,0.24	0.18	–0.06	–0.26,0.14	0.57
Mother’s BMI				0.04	–0.03,0.12	0.27				0.10	–0.01,0.21	0.09
Father’s BMI				0.03	–0.07,0.14	0.52				0.10	–0.05,0.25	0.18
Anxiety symptoms: SCARED^d^ score	Child BMI	–0.25	–0.42,0.08	0.004	–0.34	−0.59,–0.10	0.005	–0.1	–0.24,0.04	0.16	–0.24	−0.43,–0.05	0.01
Mother’s BMI				0.03	–0.05,0.10	0.48				0.12	0.01,0.23	0.03
Father’s BMI				0.04	–0.06,0.14	0.42				0.05	–0.09,0.19	0.48
ADHD symptoms: RS-DBD^e^ score	Child BMI	0.48	0.31,0.65	<0.001	0.69	0.43,0.95	<0.001	–0.09	–0.23,0.06	0.24	–0.15	–0.35,0.05	0.15
Mother’s BMI				–0.05	–0.13,0.03	0.19				0.07	–0.04,0.18	0.24
Father’s BMI				–0.10	–0.20,0.00	0.06				0.01	–0.13,0.15	0.93
ADHD-inattention symptoms: RS-DBD^e^ score, inattention items	Child BMI	0.44	0.26,0.61	<0.001	0.65	0.39,0.92	<0.001	–0.07	–0.21,0.08	0.37	–0.15	–0.35,0.05	0.15
Mother’s BMI				–0.06	–0.14,0.02	0.13				0.10	–0.01,0.21	0.09
Father’s BMI				–0.09	–0.19,0.01	0.08				–0.01	–0.15,0.14	0.93
ADHD-hyperactivity symptoms: RS-DBD^e^ score, hyperactivity items	Child BMI	0.42	0.25,0.60	<0.001	0.59	0.32,0.85	<0.001	–0.09	–0.24,0.06	0.24	–0.12	–0.33,0.08	0.25
Mother’s BMI				–0.03	–0.11,0.05	0.42				0.02	–0.09,0.13	0.70
Father’s BMI				–0.08	–0.19,0.02	0.12				0.02	–0.12,0.16	0.80
^a^Coefficients represent change in symptoms, log-transformed after adding 1, per 5 kg/m^2^ increase in BMI. ^b^Genetic models adjust for child’s sex, genotyping batch, and the first 20 principal components of ancestry. ^c^Short Mood and Feelings Questionnaire. ^d^Screen for Child Anxiety Related Disorders. ^e^Parent/Teacher Rating Scale for Disruptive Behaviour Disorders.

As the reviewer points out, more precisely null results using the full versions of the PGS would further support the hypothesis of an influence of family-level processes on the classic MR results. This is not what we see: for all phenotypes, within-family estimates using the full PGS are further from the null. However, it is unclear exactly what this reflects, because PGSs with a more liberal p-value threshold also capture more pleiotropic effects and demographic factors (i.e., effects on outcomes which are independent of BMI). This may be especially the case for anxiety, where SNPs with sub-GWS associations with BMI from both GWAS appear to have negative effects on anxiety. This is in some sense an interesting finding, but it does not shed extra light on the questions which are the focus of this paper. For this reason, we present the additional results in Author response table 1 for the reviewers but have not included them in the manuscript.

3) Given the impressive sample size of genotyped trios, the fact that the estimates for within-family MR are still rather imprecise deserves some discussion. Does within-family MR have data requirements that currently cannot be met outside of certain privately held datasets?

At 40,949 full trios (the analytic sample in the updated analysis) MoBa is currently the largest individual study in which this approach can be applied. However, new genetic data from family-based studies are becoming available, allowing analyses of this kind within and across studies, such as through the Within Family Consortium https://www.withinfamilyconsortium.com/home/. This analysis explores application of within-family MR, with the hope that it will be applied in larger data sources (including via meta-analysis) as more data becomes available.

The following line has been added to the Discussion section: ‘MoBa is currently the largest individual study in which this approach can be applied, but new data are becoming available which will allow analyses of this kind within and across studies, such as through the Within Family Consortium https://www.withinfamilyconsortium.com/home.’

Reviewer #2 (Recommendations for the authors):In addition to the broader comment about the need to demonstrate that the study would be powered to detect an association if one is present, I have only one other substantive scientific comment:1. The multivariate regression analyses control for a series of parental traits, while the MR analyses do not. Meanwhile, the MR analyses control for genetic ancestry (in the form of PCs), while the regression analyses do not. Could the difference in results between multivariate regression and MR come from the difference in the covariates used, rather than the use of the genetic instrument? Is there a good reason to believe that controlling for ancestry would make no difference in the regression analysis, or that controlling for parental traits would make no difference in the MR analysis?

Controlling for parental traits such as depressive symptoms in genetic analyses may lead to biased estimates, as these may be colliders of associations between parental genotype and offspring phenotypes. However, to increase comparability of the models, we have updated non-genetic models and classic MR models to include all covariates controlled for in within-family models: thus, all models now adjust for the child’s, mother’s, and father’s principal components of ancestry, genotyping centre, and chip. This did not materially affect our results or conclusions. The results from the non-genetic models with and without adjustment for genetic covariates is shown in Author response table 2.

**Author response table 2. sa2table2:** 

Non-genetic associations of BMI and symptoms of depression, anxiety, and ADHD at age 8
		Full adjustment^b^	Less adjustment^c^
		Beta	CI	p	Beta	CI	p
Depressive symptoms: SMFQ^d^ score	Child BMI^a^	0.05	0.01,0.09	0.02	0.05	0.01,0.09	0.02
Mother’s BMI	0.05	0.03,0.07	<0.001	0.05	0.03,0.07	<0.001
Father’s BMI	0.01	–0.02,0.03	0.67	0.01	–0.02,0.03	0.58
Anxiety symptoms: SCARED^e^ score	Child BMI	–0.07	−0.11,–0.03	0.001	–0.07	−0.11,–0.03	<0.001
Mother’s BMI	0.01	–0.01,0.03	0.47	0.01	–0.01,0.03	0.45
Father’s BMI	–0.00	–0.02,0.02	0.89	–0.00	–0.02,0.02	0.91
ADHD symptoms: RS-DBD^f^ score	Child BMI	–0.07	−0.11,–0.03	0.001	–0.07	−0.10,–0.03	0.001
Mother’s BMI	0.04	0.02,0.06	<0.001	0.04	0.02,0.06	<0.001
Father’s BMI	0.02	–0.00,0.04	0.10	0.02	–0.00,0.05	0.07
^a^Coefficients represent S.D. change in symptoms per 5 kg/m^2^ increase in BMI. ^b^Adjusted for child’s sex, mother’s and father’s educational qualifications, mother’s and father’s depressive/anxiety symptoms and ADHD symptoms, mother’s and father’s smoking status during pregnancy, and maternal parity, and the child’s, mother’s, and father’s genotyping centre, genotyping chip, and first 20 principal components of ancestry. ^c^Adjusted for the child’s sex, mother’s and father’s educational qualifications, mother’s and father’s depressive/anxiety symptoms and ADHD symptoms, mother’s and father’s smoking status during pregnancy, and maternal parity. ^d^Short Mood and Feelings Questionnaire. ^e^Screen for Child Anxiety Related Disorders. ^f^Parent/Teacher Rating Scale for Disruptive Behaviour Disorders

In addition, I have a few comments about presentation that would make the paper easier to read and follow:2. I find it confusing the use of the phrase "multivariate regression" to describe the observational epidemiology approach, because many MR methods are arguably a form of regression themselves, just using the genetic instrument as an independent variable rather than the observed trait.

In all tables, figures, and discussion of results, we have now changed the phrase ‘multivariable regression’ to ‘non-genetic models’.

3. The Results section reads as though some pieces of it may have been reordered or moved down to the Methods section. This is most glaring at the very beginning, where the MoBa cohort and the analytic sample are not properly introduced. It would benefit from a readthrough to make sure that everything mentioned in the Results section is sufficiently explained there.

On the request of the editors, the Methods section has been moved so that it now comes before the results, in line with the standard article format for this journal. Consequently, a full description of the MoBa cohort and the analytic sample is now given before any of the results.

4. Most of the Results paragraphs read like listing off statistics, and it's very difficult to tell what conclusions we are meant to draw from each paragraph. Simply adding a concluding sentence to each paragraph would go a long way towards fixing this. It also might be helpful to see p-values in the Results, rather than just estimates and confidence intervals.

The Results section has been updated, and the following three lines added to the paragraphs summarizing results for depressive symptoms, anxiety symptoms, and ADHD symptoms:

“In summary, evidence for an effect of childhood BMI on depressive symptoms was strongest using the genetic variants for adult BMI.”

“In summary, there was little evidence from any genetic model that childhood BMI affects anxiety symptoms.”

“Thus, as for depressive symptoms, evidence for an effect of childhood BMI on ADHD symptoms was inconsistent and only detected using the adult BMI polygenic score.”

We have also added p values to all tables.

5. The MR methods used are not described in very much detail, and the MR methods papers cited in the introduction are not sufficient for me to reconstruct the analyses performed. At a minimum, the methods used in the primary analysis in figures 2-3 and tables 2-3 should be identified by name like the methods used in the robustness analysis in tables 5-6. Ideally the methods and the differences between them should also be described in more detail in the text, as not all readers will necessarily be familiar with the range of MR methods available.

The statistical analysis part of the methods section has been updated to clarify that both classic MR and within-family MR models were performed using two-stage least-squares instrumental variable regression with Stata’s ivregress, as well as specifying details e.g. regarding covariates. Further detail is meanwhile provided in Appendix 1, where the model equations are specified.

The relevant section of the methods now reads:

‘All MR models were conducted with two-stage least squares instrumental-variable regression using Stata’s ivregress, with F-statistics and R^2^ values obtained using ivreg2. Classic MR models, which do not account for parental genotype, used the child’s own PGS but not those of the parents to instrument the child’s BMI. Within-family MR models were multivariable MR models, in which we used PGSs for all members of a child-mother-child trio to instrument the BMI of all three individuals (model equations are provided in Appendix 1). Classic and within-family MR models were adjusted for the child’s sex and year of birth, and the genotyping centre, genotyping chip, and the first 20 principal components of ancestry for the child, mother, and father. Given skew in outcomes variables, all models used robust standard errors (Stata’s vce option) and thus made no assumptions about the distribution of outcomes. We report two sets of results, in which either the adult BMI GWAS, or the childhood body size GWAS, was used to create the BMI PGS for the child, mother, and father.’

Similarly, the end of the last paragraph of the introduction has been altered to introduce the models. This now reads:

‘To investigate this, we used a ‘within-family’ Mendelian randomization (within-family MR) design. This approach uses the child’s, mother’s, and father’s genotype data as instruments for the BMI of the child, mother, and father. Within family Mendelian randomization estimates of the effect of the child’s BMI on the outcomes are robust to demographic and family-level biases. We compared within-family MR estimates with estimates from multivariable regression of the child’s outcomes on the child’s, mother’s and father’s reported BMI, and with estimates from ‘classic’ Mendelian randomization (classic MR), in which the child’s genotype data was used to instrument the child’s BMI without controlling for the parents’ genotype.’